# VLM4VLA: Revisiting Vision-Language-Models in Vision-Language-Action Models

**Jianke Zhang[1], Xiaoyu Chen[1], Yanjiang Guo[1], Yucheng Hu[1], Jianyu Chen[1]**
[1] Institute for Interdisciplinary Information Sciences, Tsinghua University

## Abstract

Vision-Language-Action (VLA) models, which integrate pretrained large Vision-Language Models (VLMs) into their policy backbone, are gaining significant attention for their promising generalization capabilities. This paper revisits a fundamental yet seldom systematically studied question: how VLM choice and competence translate to downstream VLA policies performance? We introduce **VLM4VLA**, a minimal adaptation pipeline that converts general-purpose VLMs into VLA policies using only a small set of new learnable parameters for fair and efficient comparison. Despite its simplicity, VLM4VLA proves surprisingly competitive with more sophisticated network designs. Through extensive empirical studies on various downstream tasks across three benchmarks, we find that while VLM initialization offers a consistent benefit over training from scratch, a VLM's general capabilities are poor predictors of its downstream task performance. This challenges common assumptions, indicating that standard VLM competence is necessary but insufficient for effective embodied control. We further investigate the impact of specific embodied capabilities by fine-tuning VLMs on seven auxiliary embodied tasks (e.g., embodied QA, visual pointing, depth estimation). Contrary to intuition, improving a VLM's performance on specific embodied skills does not guarantee better downstream control performance. Finally, modality-level ablations identify the visual module in VLM, rather than the language component, as the primary performance bottleneck. We demonstrate that injecting control-relevant supervision into the vision encoder of the VLM yields consistent gains, even when the encoder remains frozen during downstream fine-tuning. This isolates a persistent domain gap between current VLM pretraining objectives and the requirements of embodied action-planning. Project Page: https://cladernyjorn.github.io/VLM4VLA.github.io.

## 1 Introduction

Vision-Language-Action (VLA) models (Brohan et al., 2023) have recently emerged as a central research focus, as they leverage the extensive visual-language knowledge from Vision-Language Models (VLMs) as a prior for enhancing the generalization of robotic strategies (Kim et al., 2024; Black et al., 2024; Zhang et al., 2024; Chen et al., 2025c). The majority of existing VLA methods have focused on developing more advanced network architectures (Li et al., 2023; Shi et al., 2025), incorporating additional training paradigms or modalities (Zheng et al., 2024; Chen et al., 2024a; Zhang et al., 2025), and refining action decoding schemes (Zhao et al., 2023; Pertsch et al., 2025; Wen et al., 2025). However, limited attention (Liu et al., 2025) has been given to a fundamental question at the core of VLA: how do the choice and specific capabilities of the underlying VLM affect the performance of VLA policies?

In this paper, we revisit this crucial problem. To provide a fair and clean test interface that evaluates the capabilities of VLMs without introducing extraneous variables, we first build the generic **VLM4VLA** pipeline to convert general-purpose VLMs into VLAs, as shown in Figure 2. VLM4VLA is a carefully designed network plug-in, introducing fewer than 1% new parameters. To enhance the stability of inference and the robustness of evaluation, we use a simple MLP head rather than a diffusion-based approach, thus controlling stochasticity and reducing tuning complexity. This network allows us to train the modified VLMs directly using downstream robot data, facilitating alignment between

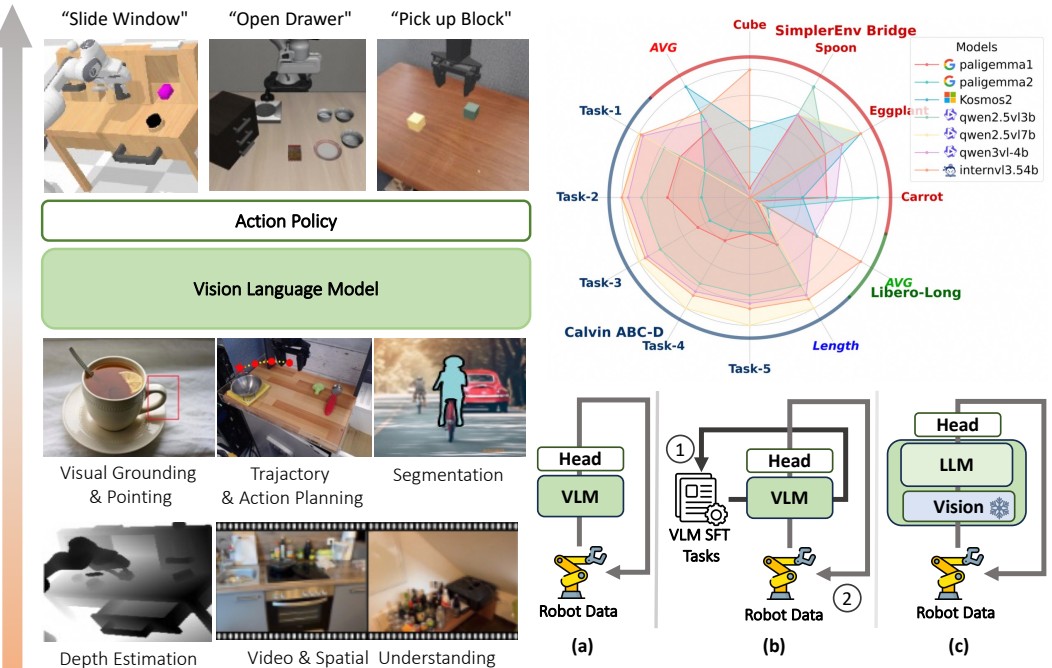

Figure 1: An overview of our **VLM4VLA** framework. **(Left)** The evaluation pipeline for testing different VLM backbones, which are evaluated on downstream tasks after an optional fine-tuning stage on auxiliary embodied tasks. **(Bottom Right)** We systematically investigate three factors influencing VLM-to-VLA transfer: the choice of VLM backbone, the impact of fine-tuning on auxiliary embodied tasks, and the vision encoder's training strategy (frozen vs. fine-tuned). **(Top Right)** A visualization of inconsistent performance of various VLM backbones across downstream tasks.

the VLM's capabilities and the demands of robotic tasks. Despite its simplicity, VLM4VLA proves effective, demonstrating competitive performance against more advanced network designs, such as the flow-matching action expert (Black et al., 2024), on benchmark tests. This provides a solid foundation for conducting fair and scalable experiments across various VLMs, aligning their capabilities with the demands of embodied tasks.

Based on the **VLM4VLA** pipeline, we conduct a large-scale empirical study across various downstream tasks on three commonly used benchmarks, evaluating 17 different VLMs that are either zero-shot or fine-tuned. Specifically, we investigate the VLM capabilities across three dimensions:

- **General capability.** Given that the VLM backbone provides superior generalization in VLAs, we first explore the relationship between the general capabilities of VLMs and their performance on downstream control tasks.

- **Embodied-specific capability.** Recent works have sought to enhance VLAs by improving the VLM backbone on a series of VLM tasks. We examine how the embodied-specific capabilities of VLMs correlate with performance on downstream control tasks.

- **Vision encoder.** Finally, we investigate the influence of the vision encoder in VLMs on downstream control task performance.

For general capability, we select seven open-sourced VLMs as the backbone. As shown in Figure 1 (a), we apply the VLM4VLA pipeline directly to these VLMs and fine-tune them with robot data. For embodied-specific capability, we collect seven commonly used embodied tasks or pretrained models based on the `Qwen2.5-VL` (Bai et al., 2025) backbone to conduct ablation studies. As shown in Figure 1 (b), we either fine-tune the VLM using auxiliary datasets or directly use the released fine-tuned models. We then apply the VLM4VLA pipeline to adapt these models to the embodied policies. Lastly, for the vision encoder, we test three VLMs with either frozen or fine-tuned vision encoders (Figure 1 (c)). We evaluate the fine-tuned models on a range of tasks across three

benchmarks including Calvin (Mees et al., 2022), SimplerEnv (Li et al., 2024), and Libero (Liu et al., 2023) to assess their embodied capabilities.

To our suprise, our findings reveal that a VLM's general capabilities are poor predictors of its downstream task performance, contrary to common assumptions. For instance, we observe that Kosmos (Peng et al., 2023) outperforms `Qwen-2.5VL` (Bai et al., 2025) and `Paligemma` (Steiner et al., 2024) across multiple environments. Inconsistencies across benchmarks suggest that VLA policies require capabilities beyond those currently pursued by VLMs. Furthermore, the gains by fine-tuning VLMs on specific auxiliary embodied tasks do not transfer to the downstream control tasks. Lastly, we identify the vision encoder as a critical bottleneck, with fine-tuning the vision encoder proving crucial for strong control performance.

Taken together, these results indicate a significant gap between current VLM research and the practical demands of VLA models. We believe this work highlights the fundamental question of the VLM's role in embodied agents and can help guide future exploration in this area.

## 2 RELATED WORKS

**Vision-Language-Action Models**   Recent works are increasingly focusing on introducing Vision-Language-Models (VLMs) (Dai et al., 2024; Touvron et al., 2023; Wang et al., 2025; Bai et al., 2025) into robot policies to enhance their generalization capabilities (Brohan et al., 2023; Guo et al., 2025; Hu et al., 2024; Chen et al., 2024b), which are known as Vision-Language-Action (VLA) models. Early methods like `RT-2` (Brohan et al., 2023) and `OpenVLA` (Kim et al., 2024) discretized actions into language tokens, enabling action learning through the VLM's autoregressive framework. Subsequent works have utilized policy heads to decode continuous actions, gradually evolving the VLA design into a hierarchical structure that combines a VLM with a policy head (Black et al., 2024; Zhang et al., 2024; Cui et al., 2025). However, most prior works focused on constructing complex policy networks, while overlooking the impact of the VLM backbone itself on the overall VLA performance. Though `RoboVLMs` (Liu et al., 2025) compared the influence of a few early VLM backbones, it did not ensure consistency in its implementations. In contrast, we design a fairer experimental framework to comprehensively test the impact of various advanced VLMs across multiple environments. By employing the simplest and a consistent additional action policy, we minimize the influence of the policy head component on the experimental results.

**Embodied Tasks for VLM**   Many prior works have explored introducing embodiment-related tasks into Vision-Language Models to enhance their spatial understanding (Yang et al., 2025; Feng, 2025) and task execution capabilities (Intelligence et al., 2025; Zhou et al., 2025b). These tasks include dense prediction (Zhen et al., 2024) and autoregressive VQA tasks (Rana et al., 2023; Yuan et al., 2024; Sermanet et al., 2024). Recently, inspired by the hierarchical System 1 and System 2 design paradigm (Zhang et al., 2024; Shentu et al., 2024), much research focused on using VLMs to build a general-purpose "robot brain"—a large language model capable of planning or completing real-world tasks via language. For instance, `Robo2vlm` (Chen et al., 2025b) annotated datasets like Open-X for VQA tasks, while `Robobrain` (Team, 2025) constructed affordance and visual trace tasks from `RoboVQA` (Sermanet et al., 2024) and video datasets. Additionally, many VLA studies leverage auxiliary post-training or co-training tasks of VLM (Intelligence et al., 2025; Zhang et al., 2025) to obtain better backbone networks. In our subsequent experiments, we fine-tune VLM on various post-training tasks and compare its performance as a VLA backbone against the corresponding VLM baselines, thereby validating the effectiveness of these auxiliary tasks for VLA performance.

## 3 STUDY DESIGN

To create a comprehensive and fair benchmark of VLM performance on manipulation tasks, we designed the VLM4VLA framework around three core principles:

- **Fairness and Reproducibility**   We employ a consistent model architecture and training/testing settings across multiple simulation environments to ensure fair and reproducible comparisons.
- **Minimalist Design**   We encapsulate VLMs within a simple yet effective VLA framework, thereby minimizing the influence of complex, extraneous policy designs on the comparison.

- **Isolating Core VLM Capabilities** To directly evaluate the VLM's intrinsic knowledge, our framework relies exclusively on visual-language inputs. We format input sequences to match each VLM's native instruction-tuning format and deliberately exclude other modalities like proprioceptive state or tactile feedback. This isolates the VLM's contribution and directly tests how its intrinsic capabilities translate to manipulation.

In the following sections, we will detail the VLM4VLA research framework. In Sec 3.1 we introduce the basic experimental design. We then describe the model architecture within VLM4VLA in Sec 3.2, and Sec 3.3 explains the experiment setup and evaluation protocol.

## 3.1 BASIC EXPERIMENT DESIGN

As shown in Figure 1, our objective is to investigate the impact of different VLMs, as well as VLM-specific fine-tuning tasks, on the performance of the resulting VLA. Our experiments primarily focus on VLMs ranging from 1B to 10B parameters and various embodiment-related auxiliary tasks. Beyond fundamental vision-language alignment, the general capabilities of VLMs relevant to embodied tasks include visual grounding, trajectory and action prediction, task planning, video understanding, and spatial reasoning. Some models also possess image rendering capabilities, such as depth prediction and semantic segmentation, as seen in works like Beyer et al. (2024); Chen et al. (2025a); Deng et al. (2025). To compare the effectiveness of these models and tasks, we encapsulate each VLM—both in its original form and after being fine-tuned on a specific auxiliary task—into a VLA, following the methodology described in Sec 3.2. All VLAs, despite their different backbones, share an identical action encoding and decoding scheme and introduce a minimal number of additional trainable parameters (less than 1%).

Subsequently, these VLM4VLA models are trained on robotic datasets using a consistent training setup. During this process, we fine-tune all model parameters, as our studys in Sec 4.3 demonstrate that freezing parts of the model leads to significant performance degradation. The fine-tuned models are then deployed for rollouts in the target environments, where we measure the success rates on various tasks. The VLA performance is assessed according to the protocol detailed in Sect 3.3.

## 3.2 VLM4VLA NETWORK DESIGN

In this section, we detail the method for constructing a consistent VLA from various VLMs within the **VLM4VLA** framework, as illustrated in Figure 2. Our objective is to build a VLA architecture that is generic across different VLMs, lightweight, and capable of fully leveraging the VLM's intrinsic knowledge.

We introduce a learnable action query token to extract embodiment-related knowledge from the VLM. The representation of this token is then decoded into an action chunk. To align with the pre-training input format of each model, we adapt a unique token concatenation scheme for each VLM4VLA instance. We take the

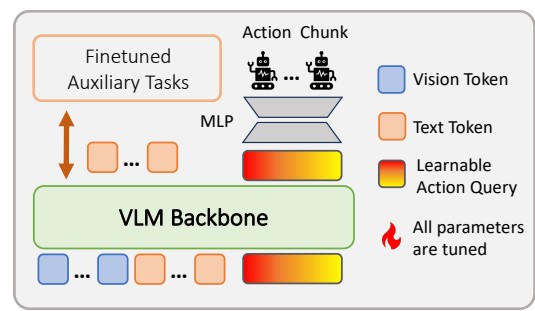

Figure 2: VLA Network in VLM4VLA

last_hidden_state of the ⟨ActionQuery⟩ token, as encoded by the VLM, and decode it into an action chunk using a small MLP-based policy head. The overall action output can be formulated as:

$$\mathbf{action} = \mathrm{MLP}(\mathrm{VLM}([\langle img \rangle \dots \langle img \rangle \langle text \rangle \dots \langle text \rangle \langle ActionQuery \rangle)]))$$

where ⟨img⟩ represents the visual embeddings from the vision encoder, ⟨text⟩ corresponds to the language embeddings containing the instruction and any additional prompts, and ⟨ActionQuery⟩ is the learnable action query token. The token sequence above omits VLM-specific special tokens (e.g. ⟨PAD⟩ or ⟨EOS⟩ ...) for brevity. Further details can be found in Appendix A.2.

**Training Objective** During training, we finetune all parameters of the VLM, including the LLM, the vision encoder, and the word embeddings. We deliberately avoid widely-used objectives like diffusion loss and flow-matching loss. Our preliminary experiments revealed that these losses

introduce significant stochasticity during inference, requiring a much larger number of rollouts for accurate evaluation. They also cause substantial performance fluctuations between different checkpoints late in training, which is not conducive to the fair comparison we aim to achieve. As a result, we utilize a maximum likelihood imitation learning objective. The desired relative position $\boldsymbol{a}^{\text{pos}}$ of the end-effector (or continuous joint action) is optimized via a MSE loss. The discrete status $a^{\text{end}}$ of the end-effector is optimized with a binary cross-entropy loss:

$$\mathcal{L} = \frac{1}{|\mathcal{B}|} \sum_{\mathcal{B}} \left( \|\boldsymbol{a}^{\text{pos}} - \hat{\boldsymbol{a}}^{\text{pos}}\|_2^2 + \text{BCE}(a^{\text{end}}, \hat{a}^{\text{end}}) \right) \tag{1}$$

where $\hat{\boldsymbol{a}}^{\text{pos}}$ and $\hat{a}^{\text{end}}$ denote the demonstration data for the relative position and status of the end-effector in a sampled mini-batch $\mathcal{B}$.

### 3.3 EXPERIMENT SETTINGS AND EVALUATION PROTOCOL

We fine-tune the VLM and the action policy for each model using identical hyperparameters and model configurations for fair comparisons. We conduct a learning-rate sweep and select a unified hyperparameter configuration to ensure that all models reached convergence at evaluation time (details can be found in Appendix A.2. Specifically, the model uses a single-view image of the current frame as its visual input and does not take proprioceptive information; this prevents the model from directly learning actions from the state. To handle inconsistent input image sizes across different VLMs, we standardize the input to $224 \times 224$ resolution for all training. If a VLM requires a different input size, we first process the image at $224 \times 224$ and then resize it to the model-specific dimensions. During training, all parameters of VLM are trained, including the vision encoder, word embeddings, LLM, and small policy head. For each VLM, we use a simple instruction prompt consistent with its pre-training format (detailed prompts format for each model can be found in Appendix A.2.3).

To ensure the reproducibility and fairness of our experiments, we do not use real-world tasks for evaluation. A direct consequence of this choice is that it prevents other researchers from directly comparing their own models against our findings on physical robots. We test different models in three simulation environments: Calvin (Mees et al., 2022), SimplerEnv (Li et al., 2024), and Libero (Liu et al., 2023). Since each environment requires separate training and testing, to improve the efficiency and rigor of our evaluation, we select the most challenging scenarios as our evaluation benchmarks. During tests, we experiment with executing the full action chunk, half of the action chunk, and a single step over all validation checkpoints. We report the best-performing result. More details about the settings of different environments can be found in the Appendix A.1.

**Calvin ABC-D**  We evaluate on the Calvin ABC-D task. We train the model for 30k steps on the ABC splits and evaluate it on 1000 task sequences, each with a length of 5. During testing, the policy is required to complete a sequence of 1–5 tasks. This setup challenges the VLM's ability to generalize to novel visual scenes. We report the average number of successfully completed tasks per sequence.

**SimplerEnv Bridge**  To better differentiate the performance of various VLM-based policies, we choose the WindowX (Bridge V2) task suite, which is more challenging than the Fractal suite. We train for 50k steps on `Bridge-V2` (Walke et al., 2023). During evaluation, we run 24 trials with random initializations for each of the four scenes (`Pick Carrot`, `Pick Eggplant`, `Pick Spoon`, and `Stack Cube`) and calculate the success rate.

**Libero-Long (-10)**  Among the five task suites in Libero, we evaluate different models on the most challenging suite `Libero-Long`, which consists of 10 tasks involving a variety of objects and skills. All models are trained for 50k steps on the training split and evaluated with 50 trials with random initializations for each task.

## 4 EXPERIMENT AND RESULTS

To comprehensively evaluate how VLM capabilities transfer to robot manipulation, our experimental analysis is three-fold. First, in Sec 4.1, we benchmark various open-source VLMs on a challenging set of tasks across three simulators to assess the relationship between general capability and the

downstream performance. Second, in Sec 4.2, we investigate whether improvements gained from fine-tuning on specific auxiliary embodied tasks translate to better final performance. Finally, in Sec 4.3, we analyze the importance of the vision encoder by comparing the outcomes of freezing versus fine-tuning it during VLA adaptation.

## 4.1 Performance of Different VLMs on VLM4VLA

### 4.1.1 Baselines

**VLM Baselines** We evaluate several Vision-Language Models commonly used in open-source VLAs, with model sizes generally ranging from 1B to 10B parameters, which ensures the feasibility of extensive action-learning finetuning and rollout testing. We test a variety of VLM models from the VLA domain, including the Paligemma series (`paligemma-1` (Beyer et al., 2024) and `paligemma-2` (Steiner et al., 2024)), the QwenVL series (`Qwen2.5VL-3B`, `Qwen2.5VL-7B` and `Qwen3VL-4B`) (Bai et al., 2025), InternVL3.5-4B (Wang et al., 2025) and Kosmos-2 (Peng et al., 2023). Among these, the `QwenVL` and `InternVL` series are top-tier general-purpose open-source VLMs, `Paligemma` is designed for better adaptability to downstream finetuning, and the `Kosmos` series excels at grounding tasks. These models cover a range of multimodal LLMs with diverse architectures and strengths, allowing us to comprehensively compare their action-learning capabilities.

**VLA Baselines** For comparison, we select several expert VLAs as reference baselines, which use different VLMs as their backbones:

- `OpenVLA` (Kim et al., 2024): Uses `Llama2-7B` with `DINOv2/SigLIP` as its VLM backbone. It differs from our VLM4VLA framework by decoding actions into a discrete space. The total model size is approximately 7.7B. For the Calvin environment, we report our reproduced OpenVLA results, while for the Simpler and Libero environments, we report the official results. Similar to our VLM4VLA setup, this model uses only a single image as input and does not leverage proprioceptive state.

- `pi0` (Black et al., 2024): Based on `Paligemma-1`, with a total size of approximately 3.1B. We modify the code provided by *open-pi-zero* to train and test within our setups, ensuring consistent settings with other backbones. We remove the proprioceptive expert and use a single image as input. More details can be found in Appendix A.3.

- `ThinkAct` (Huang et al., 2025a): A recent VLA model enhanced with reinforcement learning, based on `Qwen2.5VL-7B`. We report its official results on Simpler and Libero for comparison against our `Qwen2.5VL-7B` baseline. Unlike other baselines, `ThinkAct` takes use of proprioceptive state as an input.

### 4.1.2 Main Results

The performance of different VLMs on robotics tasks is presented in Tables 1 and 2. In addition to success rates, we report the number of finetuned parameters for each model in the *size* column.

As shown in Table 1 for the Calvin ABC-D task, `QwenVL` and `InternVL` significantly outperform other VLMs. Notably, the average of 4.057 completed tasks by `Qwen2.5VL-7B` is nearly on par with state-of-the-art VLAs. Furthermore, VLMs that perform better on QA-benchmarks, such as the `QwenVL` series, also exhibit superior performance on Calvin, suggesting a correlation between the capabilities tested by Calvin and other VQA benchmarks. Concurrently, we observe that `pi0`, which is based on `Paligemma-1` and does not use state information, performs similarly to the base `Paligemma-1` model. This indicates that the additional action expert is constrained by the inherent capabilities of VLM backbone itself and fails to yield performance improvements in the Calvin environment. Table 2 displays the performance of the models on the Simpler-Bridge and Libero-10 (long) tasks. We find that `KosMos-2`, the smallest model, achieves the highest success rate on both tasks. On the Simpler-Bridge task, the `Paligemma` series models outperform the `Qwen2.5VL` series, whereas on Libero-10, the performance across different VLM categories is comparable. It is worth noting that `ThinkAct`, finetuned from `Qwen2.5VL-7B`, performs similarly to the base `Qwen2.5VL-7B` on Simpler-Bridge. However, its performance on Libero-10 is substantially better than all other models. This may be caused by the proprioceptive state information utilized by `ThinkAct`, which can be a useful input for Libero environment.

Table 1: Results on Calvin ABC-D. Entries marked with * are expert VLAs modified and reproduced with our training and test settings.

| Model (VLM Backbone) | Size | Task-1 | Task-2 | Task-3 | Task-4 | Task-5 | Calvin↑ |
|---|---|---|---|---|---|---|---|
| *Expert Vision-Lanugage-Action Models* | | | | | | | |
| OpenVLA* (Llama-2) | 7.7B | 0.792 | 0.644 | 0.499 | 0.368 | 0.245 | 2.548 |
| pi0* (Paligemma-1) | 3.1B | 0.896 | 0.785 | 0.786 | 0.610 | 0.532 | 3.509 |
| *VLM with VLM4VLA Models* | | | | | | | |
| Qwen2.5VL-3B | 3.8B | 0.922 | 0.842 | 0.766 | 0.700 | 0.626 | 3.856 |
| Qwen2.5VL-7B | 8.3B | 0.935 | 0.864 | 0.807 | 0.758 | 0.693 | 4.057 |
| Qwen3VL-4B | 4.4B | 0.933 | 0.857 | 0.790 | 0.719 | 0.644 | 3.943 |
| Paligemma-1 | 2.9B | 0.914 | 0.813 | 0.692 | 0.599 | 0.488 | 3.506 |
| Paligemma-2 | 3.0B | 0.901 | 0.775 | 0.669 | 0.575 | 0.486 | 3.406 |
| KosMos-2 | 1.7B | 0.878 | 0.721 | 0.591 | 0.498 | 0.408 | 3.096 |
| InternVL3.5-4B | 4.7B | 0.934 | 0.864 | 0.795 | 0.728 | 0.656 | 3.977 |

Table 2: Results on SimplerEnv-Bridge and Libero-10. Entries marked with * are expert VLAs modified and reproduced with our training and test settings.

| Model (VLM Backbone) | Size | Carrot | Eggplant | Spoon | Cube | Simpler↑ | Libero↑ |
|---|---|---|---|---|---|---|---|
| *Expert Vision-Lanugage-Action Models* | | | | | | | |
| OpenVLA (Llama-2) | 7.7B | 4.2 | 0.0 | 0.0 | 12.5 | 4.2 | 53.7 |
| pi0* (Paligemma-1) | 3.1B | 62.5 | 100.0 | 54.2 | 25.0 | 60.4 | 46.0 |
| ThinkAct (Qwen2.5VL-7B) | 7.4B | 37.5 | 70.8 | 58.3 | 8.7 | 43.8 | 70.9 |
| *VLM with VLM4VLA Models* | | | | | | | |
| Qwen2.5VL-3B | 3.8B | 20.8 | 91.7 | 79.2 | 0.0 | 47.9 | 43.0 |
| Qwen2.5VL-7B | 8.3B | 12.5 | 100.0 | 75.0 | 0.0 | 46.9 | 45.0 |
| Qwen3VL-4B | 4.4B | 54.2 | 95.8 | 75.0 | 0.0 | 56.3 | 44.4 |
| Paligemma-1 | 2.9B | 50.0 | 91.7 | 75.0 | 4.2 | 55.2 | 44.2 |
| Paligemma-2 | 3.0B | 75.0 | 75.0 | 79.2 | 0.0 | 57.3 | 46.2 |
| KosMos-2 | 1.7B | 37.5 | 100.0 | 75.0 | 29.2 | 60.4 | 55.0 |
| InternVL3.5-4B | 4.7B | 12.5 | 100.0 | 62.5 | 54.2 | 57.3 | 62.8 |

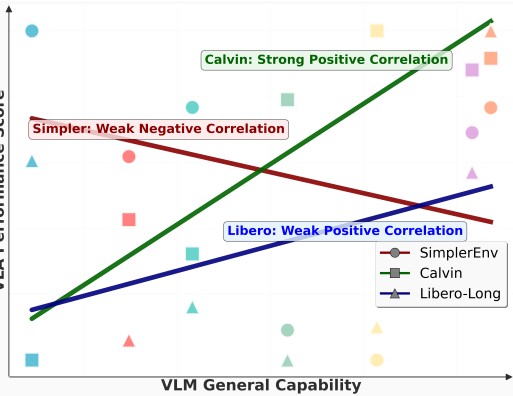

As illustrated in Figure 3, we conducted a linear regression analysis to examine the relationship between VLA performance and the general capabilities of the underlying VLMs. We plot the performance of various VLMs on several general-purpose VQA benchmarks against their performance on VLA tasks. For `Paligemma` and `Kosmos`, we approximated their general capabilities using results from their proprietary tasks. For `Qwen-VL` and `InternVL`, we used the average scores from multiple general-purpose VQA benchmarks. A linear regression line is fitted to visualize the correlation between these two sets of metrics. The results indicate that different evaluation environments exhibit varying degrees of linear correlation with these general VLM capabilities. We found that the results on the Calvin benchmark exhibit a high correlation

Figure 3: Comparison of the linear relationship between general VLM capabilities and VLA performance.

with performance on VQA benchmarks. In contrast, for the Simpler and Libero environments, there is no apparent correlation between a VLM's QA performance and its VLA performance. This suggests a significant gap exists between the capabilities required for VLA manipulation tasks and those measured by existing VQA benchmarks. More details about constructing Figure 3 can be found in Appendix A.4.

## 4.2 IMPACT OF DIFFERENT VLM AUXILIARY TASKS ON VLA PERFORMANCE

Recent works have proposed using robotic data to construct VQA datasets for improving VLM backbones, e.g., Robobrain (Team, 2025), Robix (Huang et al., 2025b). However, few studies have investigated whether this additional continual finetuning actually benefits VLAs in downstream tasks. In this section, we construct or collect several SFT tasks for VLM, including VQA datasets and a generation task. We first finetune the `Qwen2.5VL` model and subsequently, we employ each finetuned VLM as the backbone for our `VLM4VLA` framework and evaluate its performance on the Calvin benchmark. Specifically, we compare the following finetuned VLMs (more details about SFT datasets can be found in Appendix A.6):

**Robopoint** (Yuan et al., 2024) A pointing task dataset collected in simulator. Given an image and a target location, the model is required to output the 2D coordinates that satisfy the target requirement. This finetuning dataset contains 1.432M samples. After finetuning, the performance of `Qwen2.5VL-3B` on the pointing task improved by ~20% on the test split.

**Vica-332k** (Feng, 2025) A spatial understanding dataset constructed from RGB-D datasets. It covers a wide range of capabilities, including size estimation, position understanding, distance estimation, and so on. After finetuning `Qwen2.5VL-3B` on this dataset, we achieve a performance improvement of ~15% on VSI-Bench Yang et al. (2025).

**Bridgevqa** We used `VQASynth` to annotate data from Bridge-v2, Fractal, and Calvin ABC, combining them into a spatial understanding question-answering dataset. The main QA content includes judging the distance, size, and depth between two objects.

**Robo2vlm** (Chen et al., 2025b) An action-oriented question-answering dataset built from 176k real robot trajectories, containing 667k VQA pairs. It involves tasks such as multi-view understanding, success prediction, position management, and trajectory judgment. After finetuning `Qwen2.5VL-3B` on this dataset, we achieved a 60% performance improvement on its test set.

**Robobrain2** (Team, 2025) A large-scale embodied VQA dataset and a VLM finetuned on `Qwen2.5VL-7B`. The tasks include pointing, planning, and marking trajectory. We directly use their finetuned VLM as the backbone for our `VLM4VLA`.

**Omni-Generation** We integrate a diffusion model into `QwenVL-7B` and train on both image generation, depth map generation, and semantic segmentation map generation tasks together with general VQA tasks. We use the resulting VLM part as the backbone for `VLM4VLA`.

**VQA-Mix** We mix the aforementioned VQA datasets with other large-scale general data to finetune `Qwen2.5VL-7B`.

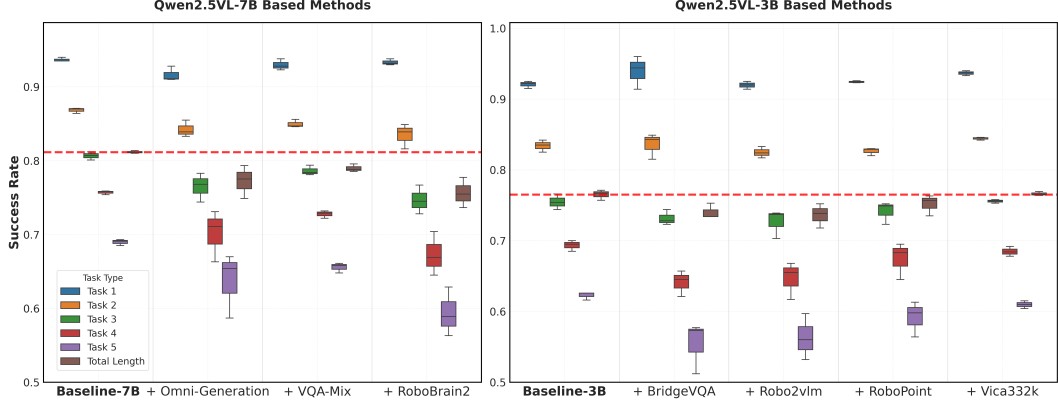

Figure 4: Performance of different auxiliary VLM finetune tasks. The 'Length' dimension is scaled by a factor of 5 to normalize it to the range [0, 1]. The results for the VLAs trained under each task and for each gradient steps (10k, 15k, 20k, 25k and 30k) are rendered as box plots to provide a view of the impact of different tasks on the VLA's performance and stability.

Figure 4 presents the performance of the various finetuned VLMs. Overall, all models underperform the original baseline, with most exhibiting a slight degradation in performance and an obvious increase in variance. For `Qwen2.5VL-3B`, the model finetuned on `Vica332k` performs better

than those finetuned on other datasets. This could be attributed to the dataset's broad data coverage and diverse task types, which may prevent the model from overfitting to a narrow set of capabilities and consequently degrading others. For `Qwen2.5VL-7B`, the `VQA-Mix` model shows the least performance degradation, achieving results nearly identical to the baseline. This suggests that general-purpose VQA data plays a crucial role during embodiment-focused finetuning, further implying that VLAs may require broad, general capabilities, beyond just embodied skills, to perform well on downstream tasks.

It is also worth noting that finetuning with generation tasks (i.e., `Omni-Generation` on `Qwen2.5VL-7B`), such as depth and semantic map prediction, did not yield performance benefits. This may indicate that simply introducing generation tasks or dense 3D-aware tasks into VLM finetuning process does not provide a tangible advantage for the VLA. Similarly, `Robobrain2`, positioned as a general-purpose embodied brain, also underperformed the baseline in our VLM4VLA tests. This suggest that existing embodied VQA-style tasks do not offer a clear benefit for training end-to-end VLAs to execute downstream manipulation tasks.

## 4.3 IMPORTANCE OF VISION ENCODER

Table 3: Influence of freezing vision encoder of VLMs

|  | Size | Calvin ABC-D↑ | SimplerBridge↑ |
|---|---|---|---|
| Qwen2.5VL-3B | 3.8B | 3.856 | 48.00 |
| + freeze vision encoder | 3.1B | 2.855 *(-1.001)* | 23.95 *(-24.05)* |
| Qwen2.5VL-7B | 8.3B | 4.057 | 46.75 |
| + freeze vision encoder | 7.6B | 2.823 *(-1.234)* | 25.50 *(-21.25)* |
| Paligemma-1 | 2.9B | 3.506 | 55.25 |
| + freeze vision encoder | 2.5B | 0.495 *(-3.011)* | 13.25 *(-42.00)* |

Table 3 shows the performance of three models when the vision encoder is frozen during `VLM4VLA` training. We observe a significant performance degradation for all models on both the Calvin and Simpler benchmarks after freezing the vision encoder, with this drop being particularly pronounced for the `Paligemma` models. In the case of `Qwen2.5VL-7B`, the vision encoder accounts for 0.7B parameters. When it is frozen, the total number of tunable parameters is still a substantial 7.6B, which is much larger than 3.8B in `Qwen2.5VL-3B`. However, the performance of the frozen `Qwen2.5VL-7B` is not only significantly worse than its fully finetuned counterpart but also substantially underperforms the fully finetuned `Qwen2.5VL-3B`. This observation is consistent with findings in prior work (Kim et al., 2024; Huang et al., 2025c).

This finding strongly suggests that finetuning the vision encoder is crucial when adapting a VLM into a VLA, and that the impact of this module can be more significant than merely increasing the number of trainable parameters in the language model. A potential reason for this phenomenon is that the VLM's pre-trained vision encoder is not well-aligned with the visual domain of embodied scenes. The model likely requires significant adaptation of its vision module to effectively process and align with the visual signals from these novel environments. This highlights a promising direction for future research.

## 4.4 INFLUENCE OF PRETRAINED VLM

To isolate the contribution of the VLM backbone, we establish a lower-bound baseline by training VLM4VLA variants entirely from random initialization. This clarifies whether the observed generalization stems from the architecture alone or from VLM pre-training. We train Qwen2.5-VL (3B and 7B) and PaliGemma from scratch on the Calvin ABC-D and Simpler-Bridge benchmarks. We strictly match the training hyperparameters and evaluation protocols reported in the main paper for all from-scratch runs.

Models trained from scratch exhibit substantial degradation across both benchmarks and all model sizes. This gap indicates that VLM pre-training is fundamental to the generalization ability of the VLA model, beyond architectural design. Training from scratch yields markedly lower perfor-

Table 4: Performance of training from scratch compared to pretrained initialization. We report absolute scores and the change (in parentheses) relative to pretrained models on Calvin ABC-D and Simpler-Bridge.

| Model | Calvin ABC-D | Simpler-Bridge |
|---|---|---|
| Qwen2.5-VL-3B (pretrained) | 3.856 | 48.00 |
| Qwen2.5-VL-3B (from scratch) | 1.381 (-2.475) | 15.75 (-32.25) |
| Qwen2.5-VL-7B (pretrained) | 4.057 | 46.75 |
| Qwen2.5-VL-7B (from scratch) | 1.769 (-2.288) | 18.20 (-28.75) |
| PaliGemma-1 (pretrained) | 3.506 | 55.25 |
| PaliGemma-1 (from scratch) | 1.129 (-2.377) | 14.50 (-40.75) |

mance, confirming that VLM pre-training is a prerequisite for strong downstream generalization in VLM4VLA.

## 5 CONCLUSIONS

In this paper, we investigated the impact of various VLMs, including the effect of auxiliary fine-tuning tasks, on the performance of VLA models. Through extensive training and evaluation experiments conducted across three distinct environments, we assessed the capabilities of seven models and seven categories of auxiliary data for executing manipulation tasks. A core insight from our study is the significant gap between the capabilities of current VLMs and the demands of VLA embodied tasks. Specifically, we observe a notable discrepancy between general VLM capabilities and its VLA effectiveness. While comparisons with training-from-scratch baselines confirm that pre-trained VLMs are a necessary foundation, their general capabilities are poor predictors of downstream success. Furthermore, even improving specific embodied skills does not guarantee better control. These findings challenge the intuition that stronger VLMs automatically yield better agents, highlighting a disconnect between current pre-training paradigms and embodied needs. The mechanisms driving this gap remain an open question for future research. Finally, a limitation of our work is the absence of experiments on physical robots. This decision was motivated by challenges related to reproducibility, as well as difficulties in ensuring test efficiency and fairness across physical hardware setups. While real-world deployment remains the ultimate goal, we believe that our comprehensive results across multiple, diverse simulation benchmarks provide valuable insights that can inspire and guide future research in this area.

## 6 ETHICS STATEMENT

The data used in the VLM4VLA is sourced exclusively from public repositories. Our contributions fully adhere to the terms of these licenses. We did not use any data beyond what is publicly available and downloadable.

## 7 REPRODUCIBILITY STATEMENT

All results and experimental conclusions in this paper are reproducible. To facilitate reproducibility, we have included our source code in the supplementary material. We are committed to open science and plan to publicly release the complete codebase, trained models, datasets, and evaluation logs upon publication of this work.

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

# A   APPENDIX

## A.1   MORE DETAILS ABOUT TESING ENVIRONMENTS

**Calvin ABC-D** Calvin is a simulation benchmark for various tabletop manipulation tasks. Its dataset consists of four splits (A, B, C, and D), each with different scene configurations (primarily changes in object and scene colors). During testing, the policy is required to complete a sequence of 1–5 tasks. To prevent larger models from merely overfitting to seen color schemes, we adopt the `ABC-D` setup, where the model is trained on scenes A, B, and C, and tested on scene D. This setup challenges the VLM's ability to generalize to novel visual scenes. We train the model for 30k steps on the ABC splits and evaluate it on 1000 task sequences, each with a length of 5. We report the average number of successfully completed tasks per sequence.

**SimplerEnv Bridge** SimplerEnv is a suite of real-to-sim evaluation environments where the policy is trained on real-world data and tested in simulation. The test scenarios are divided into two categories based on the data source: Google Robot (Fractal) and WindowX (Bridge V2). The former has a smaller visual gap between the training and testing domains, where most policies perform well. To better differentiate the performance of various VLM-based policies, we choose the more challenging Bridge split as our evaluation metric. We train for 50k steps on `BridgeV2`, with validation every 5k steps (for a total of 10 checkpoints). During evaluation, we run 24 trials with random initializations for each of the four scenes (`Pick Carrot`, `Pick Eggplant`, `Pick Spoon`, and `Stack Cube`) and calculate the success rate. We report the best result from the validation checkpoints.

**Libero-Long (-10)** Libero is a simulation benchmark for multi-category manipulation tasks on a fixed tabletop, which contains five task suites. We select the most challenging suite, `Libero-Long` (also known as `Libero-10`), which consists of 10 long-horizon tasks involving a variety of objects, scenes, and manipulation types.

## A.2   MORE IMPLEMENTATION DETAILS

### A.2.1   TRAINING SETUPS

To ensure a fair comparison, we fine-tune the Vision-Language Model (VLM) and the action policy with identical hyperparameters and model configurations in each test environment. Specifically, all our experiments are conducted on 8 NVIDIA A100 GPUs. The model uses a single-view image of the current frame as its visual input and does not take state information as input, which prevents the model from directly learning actions from the state. We fine-tune all parameters of the VLM, including the vision encoder, token embeddings, and the LLM part. The detailed training hyperparameters are as follows:

- **Calvin**: We use a batch size of 256, a learning rate of $2 \times 10^{-5}$ for all models, and an action chunk size of 10.
- **Simpler & Libero**: We use a batch size of 512, a uniform learning rate of $5 \times 10^{-5}$, and an action chunk size of 4.

### A.2.2   ABOUT CHOICE OF HYPERPARAMETERS

Here, we provide additional details on hyperparameter choices (action chunk, learning rate, and batch size) and discuss the implications of using exactly the same hyperparameters across all methods.

For the results reported in the paper, we have performed a sweep on different action chunk sizes during evaluation. We also use 5 different checkpoints to mitigate the training instability. We reported the best performance among these runs. In most cases, the best performance was achieved before the final checkpoint, indicating that the models had already converged. The protocol that using the same hyperparameters ensures that our results reflect each model's peak capability (convergence), effectively decoupling performance from training proficiency or randomness.

To verify the robustness of our Learning Rate (LR) choice, we performed a sweep using values $\{1e-5, 2e-5, 5e-5, 1e-4\}$ on representative models (Qwen2.5-VL-3B, Qwen2.5-VL-7B, and PaliGemma-1). As shown in the table below, while minor fluctuations exist, the downstream

performance remains highly stable across this range, and the relative ranking of models remains consistent.

| Learning Rate | 1e-5 | 2e-5 | 5e-5 | 1e-4 |
|---|---|---|---|---|
| Qwen2.5VL-3B | 3.844 | **3.856** | 3.832 | 3.792 |
| Qwen2.5VL-7B | 4.050 | **4.057** | 4.027 | 3.986 |
| Paligemma-1 | **3.506** | **3.506** | 3.487 | 3.492 |

Table 5: Performance across learning rates on Calvin ABC-D.

Regarding batch size, we maximized this parameter uniformly across all models and make sure this stays the same. We believe that larger batch sizes benefit all architectures by stabilizing gradients; therefore, enforcing a large, identical batch size prevents any model from being disadvantaged by gradient noise.

### A.2.3 LANGUAGE PROMPTS AND FORMAT

For fair comparison between different VLMs, we prompt all models with the simplest format of prompts while keeping consistent with the VLM's pretraining prompt format. The detailed prompts for the different VLMs used in VLM4VLA are as follows (including special tokens). For convenience, here we use the robotic instruction `pick up cube` as an example of an input encoded in this manner.

**Paligemma-1 & Paligemma-2**

The `Paligemma` model employs a `prefix-suffix` encoding scheme during its pre-training. In this scheme, the prefix part is processed with bidirectional attention, while the suffix part uses unidirectional (causal) attention for generative tasks. The two parts are demarcated by a `<bos>` token. For our implementation, we adopt the prefix encoding scheme for image tokens and the suffix encoding scheme for text tokens, as this aligns with the model's generative training objective.

$$\langle\text{img}\rangle\dots\langle\text{img}\rangle\langle\text{bos}\rangle\text{pick up cube}\backslash n\langle\text{ActionQuery}\rangle$$

**Kosmos-2**

`Kosmos` is a VLM optimized for *grounding* tasks, where its training sequences contain multiple task-specific tokens. Since the action output of a VLA is not a grounding task, we do not include these additional task tokens in our input. The expectation is that the model will automatically learn to leverage its skills acquired from other tasks during the fine-tuning process.

$$\langle\text{s}\rangle\langle\text{image}\rangle\langle\text{img}\rangle\dots\langle\text{img}\rangle\langle\text{/image}\rangle\langle\text{p}\rangle\text{pick up cube}\backslash n\langle\text{/p}\rangle\langle\text{ActionQuery}\rangle$$

**Qwen2.5VL-3B & Qwen2.5VL-7B & Qwen3VL-4B**

For the `Qwen` series of models, which are instruction-fine-tuned, general-purpose conversational models, an input format that closely mirrors their training would necessitate the inclusion of both system and user prompts. We therefore compared two distinct token concatenation schemes:

- **Scheme 1 (Full Compliance):** This approach strictly adheres to the instruction-SFT format. While this ensures maximum alignment with the model's pre-training, it results in a longer input token sequence:

  $\langle|\text{im\_start}|\rangle\text{system}\backslash n\text{You are a helpful assistant.}\langle|\text{im\_end}|\rangle\backslash n$

  $\langle|\text{im\_start}|\rangle\text{user}\backslash n\langle|\text{vision\_start}|\rangle\langle\text{img}\rangle\dots\langle\text{img}\rangle\langle|\text{vision\_end}|\rangle\text{What action should the robotic arm take to pick up cube}\langle|\text{im\_end}|\rangle\backslash n$

  $\langle|\text{im\_start}|\rangle\text{assistant}\backslash n\langle\text{ActionQuery}\rangle$

- **Scheme 2 (Simplified Consistency):** This scheme aligns with the simplified approach we used for `Kosmos` and `Paligemma`. It omits the extraneous conversational prompts (e.g., system and user roles) and includes only the most essential special tokens required for sequence construction:

  $\langle|\text{im\_start}|\rangle\langle|\text{vision\_start}|\rangle\langle\text{img}\rangle\dots\langle\text{img}\rangle\langle|\text{vision\_end}|\rangle\text{pick up cube}\backslash n\langle\text{ActionQuery}\rangle$

| | Prompt | Calvin ABC-D↑ | SimplerBridge↑ |
|---|---|---|---|
| Qwen2.5VL-3B | Long | 3.856 | 48.00 |
| Qwen2.5VL-3B | Short | 3.844 *(-0.012)* | 46.75 *(-1.25)* |
| Qwen2.5VL-7B | Long | 4.057 | 46.75 |
| Qwen2.5VL-7B | Short | 3.947 *(-0.110)* | 44.00 *(-2.75)* |

Table 6: Comparison between different prompt formats. `Long` prompt denotes Scheme 1, while the `Short` prompt denote the Scheme 2.

The comparison in Table 6 indicates that prompts perfectly matching the Supervised Fine-Tuning (SFT) format perform marginally better than their minimal counterparts. Consequently, we consistently apply the Scheme 1 prompt style to the `QwenVL` and `InternVL` series in our remaining experiments to maintain alignment with their original training formats.

**InternVL3.5-4B**

The training format of InternVL is highly similar to that of Qwen-VL, as both models utilize the same Qwen-LLM backbone. Accordingly, we adopt the same prompt format as the Qwen series, with modifications made to certain differing special tokens:

$\langle|\text{im\_start}|\rangle$system$\backslash$nYou are a helpful assistant.$\langle|\text{im\_end}|\rangle\backslash$n

$\langle|\text{im\_start}|\rangle$user$\backslash$n$\langle|\text{img}|\rangle\langle\text{img}\rangle \ldots \langle\text{img}\rangle\langle|/\text{img}|\rangle$What action should the

robotic arm take to pick up cube$\langle|\text{im\_end}|\rangle\backslash$n

$\langle|\text{im\_start}|\rangle$assistant$\backslash$n$\langle\text{ActionQuery}\rangle$

### A.2.4 IMPLEMENTATION DETAILS ABOUT VLM4VLA ARCHITECHTURE

We provide the detailed model architechture in Table 7. We maintain a consistent architecture for the action-specific modules across all models, including a fixed number of learnable tokens and a uniform hidden size for the action head. Consequently, the variation in the number of additional trainable parameters is solely dependent on the VLM's native `hidden_size`. The Policy Head architecture presented here omits the final linear layer, which maps the features to the action space (i.e., `Linear(1024, 7)`).

### A.3 IMPLEMENTATION AND EVALUATION DETAILS FOR THE PI0

**Reproducing `pi0`**    In our experiments, we reproduced the `pi0` model to serve as a strong comparative baseline. We leveraged the model implementation from the official code released by Allen but excluded its proprioceptive input module. Specifically, we integrated the `pi0` model architecture into the **VLM4VLA** data-loading and training framework for fine-tuning and evaluation. This model-only migration was a deliberate choice, as we observed that the data preprocessing strategies used in the original `pi0` implementation for the Simpler environment were inconsistent with those of OpenVLA and our **VLM4VLA** framework, thus ensuring a fairer comparison.

**Modification of the Proprioceptive State Expert**    We modified the state expert in `pi0` by directly concatenating the outputs of the VLM expert and the action expert. Since the state and action experts in the original `pi0` architecture share the same parameters, our modification only alters the model's input pathway without changing its underlying structure or parameter count.

**Evaluation Protocol**    During our experiments, we discovered that the Flow-matching loss used in `pi0` leads to significant performance instability. Specifically, multiple rollouts of the same `pi0` policy in an identical starting environment could yield success rate fluctuations as high as ±20%. In stark contrast, our **VLM4VLA** approach is deterministic under the same conditions, producing identical outcomes for each run without stochasticity. This high variance necessitated a more extensive evaluation rollouts for `pi0` to obtain a reliable estimate of its true performance. Consequently, when testing `pi0`, we performed a greater number of rollouts for each checkpoint under various settings:

| VLM | Backbone Size | Policy Head Size | Learnable Tokens | LLM Hidden Size | Policy Head |
|---|---|---|---|---|---|
| Qwen2.5VL-3B | 3755M | 6.58M | 1 | 2048 | `Linear(2048,1024)` `ReLU` `Linear(1024,1024)` |
| Qwen2.5VL-7B | 8292M | 11.69M | 1 | 3584 | `Linear(3584,1792)` `ReLU` `Linear(1792,1024)` |
| Qwen3VL-4B | 4437M | 8.02M | 1 | 2560 | `Linear(2560,1280)` `ReLU` `Linear(1280,1024)` |
| Paligemma-1 | 2923M | 6.58M | 1 | 2048 | `Linear(2048,1024)` `ReLU` `Linear(1024,1024)` |
| Paligemma-2 | 3032M | 7.27M | 1 | 2304 | `Linear(2304,1502)` `ReLU` `Linear(1502,1024)` |
| KosMos-2 | 1664M | 6.58M | 1 | 2048 | `Linear(2048,1024)` `ReLU` `Linear(1024,1024)` |
| InternVL3.5-4B | 4732M | 8.02M | 1 | 2560 | `Linear(2560,1280)` `ReLU` `Linear(1280,1024)` |

Table 7: Detailed architecture of different VLM4VLA backbones.

- Calvin: We averaged the results over 3 evaluation runs per model.
- Simpler: We performed 240 evaluation runs for each model on every task.
- Libero: We performed 250 evaluation runs for each model on every task.

## A.4 Correlation analysis between VLM capability and VLA performance

Here we provide a detailed explanation of how the data points in Figure 3 are obtained and what each graphical element represents.

- The x-axis represents the general capability of different VLMs.
- Each set of three vertically aligned points with the same color corresponds to the results of the same VLM used as the VLM4VLA backbone across three benchmarks.
- Marker shapes denote benchmarks: circles for SimplerEnv, triangles for LIBERO, and squares for CALVIN (see the legend on the right).
- The three colored lines are linear fits to the points from each benchmark: red for Simpler (fit over circle markers), blue for LIBERO (fit over triangle markers), and green for CALVIN (fit over square markers).

Figure 3 uses VLMs from data sourced from the experiments reported in Tables 1 and 2. We elaborate how we evaluate different VLMs in terms of their general capabilities. InternVL and QwenVL are general-purpose VLMs that have been evaluated across a wide range of multimodal understanding benchmarks. We take the average of their officially reported results on a set of multimodal general, reasoning, text, grounding, agentic, and embodied benchmarks as the x-axis value. This set includes: MMBench v1.1 (en), MMStar, BLINK, HallusionBench, AI2D, OCRBench, MVBench, VideoMME, MMMU, MathVista, MathVision, CountBenchQA, RefCOCO (all), GPQA, OmniSpatial, RealWorldQA, ERQA, and VSI-Bench. For PaliGemma and Kosmos, which are not trained for general-purpose tasks, we use their performance on their respective specialized

| VLM4VLA Baseline | Task-1 | Task-2 | Task-3 | Task-4 | Task-5 | ALL↑ |
|---|---|---|---|---|---|---|
| Qwen2.5VL-3B | 0.922 | 0.842 | 0.766 | 0.700 | 0.626 | 3.856 |
| + Robopoint | 0.924 | 0.829 | 0.749 | 0.683 | 0.598 | 3.783 *(-0.073)* |
| + Vica332k | 0.940 | 0.842 | 0.758 | 0.692 | 0.615 | 3.847 *(-0.009)* |
| + Bridgeqva | 0.945 | 0.843 | 0.744 | 0.657 | 0.577 | 3.765 *(-0.091)* |
| + Robo2vlm | 0.925 | 0.833 | 0.737 | 0.668 | 0.597 | 3.760 *(-0.096)* |
| Qwen2.5VL-7B | 0.935 | 0.864 | 0.807 | 0.758 | 0.693 | 4.057 |
| + Robobrain2 | 0.938 | 0.849 | 0.767 | 0.704 | 0.629 | 3.887 *(-0.170)* |
| + Omni-Generation | 0.928 | 0.833 | 0.760 | 0.711 | 0.654 | 3.876 *(-0.181)* |
| + VQA-Mix | 0.938 | 0.856 | 0.794 | 0.732 | 0.658 | 3.978 *(-0.079)* |

Table 8: Comparison of different Embodied VQA finetuning on VLM4VLA (best results on each settings)

evaluation suites as the metric. Specifically, PaliGemma is evaluated on RefCOCO (all), AI2D, CountBenchQA, and GQA; Kosmos is a grounding-focused VLM, which uses only RefCOCO results as its metric. Because the latter two models are evaluated on a subset of the tasks used for the former two (reflecting their lack of general-task capability), we apply a downweighting to their scores. This yields an approximate assessment of the "general capability of different types of VLMs" as a reference.

To quantify the strength of the linear relationship between each benchmark and VLM capability, we report the correlation metrics for the three fitted lines (Pearson correlation coefficient $r$ and $R^2$):

- Calvin: $r = 0.919, R^2 = 0.844$

- SimplerEnv: $r = -0.321, R^2 = 0.103$

- Libero: $r = 0.381, R^2 = 0.145$

### A.5 DETAILED RESULTS OF AUXILIARY TASKS

The results on each finetuned VLM on VLA tasks (Calvin ABC-D) are shown in Table 8 (corresponding to Chart 4).

### A.6 DETAILS ABOUT DATASETS USED FOR FINETUNING VLM

**Robopoint** (Yuan et al., 2024) A pointing task dataset collected in a simulator. Given an image and a target location, the model is required to output the 2D coordinates that satisfy the target requirement. This finetuning dataset contains 1.432M samples. As shown in the table, after finetuning, the performance of `Qwen2.5VL-3B` on the pointing task improved by ∼20% on the test split.

**Vica-332k** (Feng, 2025) A spatial understanding dataset constructed from RGB-D datasets. It covers a wide range of capabilities, including size estimation, position understanding, distance estimation, and so on. After finetuning `Qwen2.5VL-3B` on this dataset, we achieve a performance improvement of ∼15% on VSI-Bench Yang et al. (2025).

**Bridgevqa** We used `VQASynth` to annotate data from Bridge-v2, Fractal, and Calvin ABC, combining them into a spatial understanding question-answering dataset. The main QA content includes judging the distance, size, and depth between two objects.

**Robo2vlm** (Chen et al., 2025b) An action-oriented question-answering dataset built from 176k real robot trajectories, containing 667k VQA pairs. It involves tasks such as multi-view understanding, success prediction, position management, and trajectory judgment. After finetuning `Qwen2.5VL-3B` on this dataset, we achieved a 60% performance improvement on its test set.

**Robobrain2** (Team, 2025) A large-scale embodied VQA dataset and a VLM finetuned on `Qwen2.5VL-7B`. The tasks include pointing, planning, and marking trajectory. We directly use their finetuned VLM as the backbone for our `VLM4VLA`.

**Omni-Generation** Qwen-vlo integrates a diffusion model into VLM and trains on both image generation, depth map generation, and semantic segmentation map generation tasks. We use the resulting VLM part (`Qwen2.5VL-7B` as the backbone for `VLM4VLA`.

**VQA-Mix** We mix the aforementioned VQA datasets with other large-scale general VQA data to finetune `Qwen2.5VL-7B`. The embodied related tasks include approximately 620k image-text pairs, including Vica Feng (2025), SpaceR Ouyang et al. (2025), Embspatial Du et al. (2024) and Refspatial Zhou et al. (2025a). The general VQA data consists of 360k QA pairs.

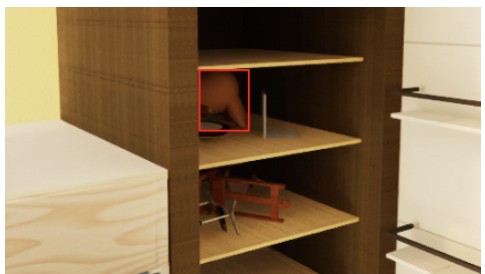

**Example of Robopoint Data**

Question: Locate several points on an item situated beneath the bordered item.

Answer: [(0.56, 0.69), (0.53, 0.76), (0.45, 0.72), (0.43, 0.67)]

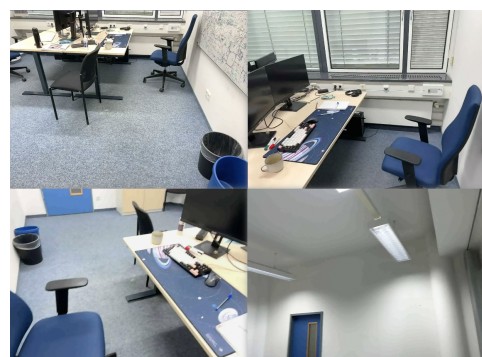

**Example of Vica Data**

Question: How large is this room in terms of square meters? If more than one room is shown, approximate the overall space.
Answer: 18.33

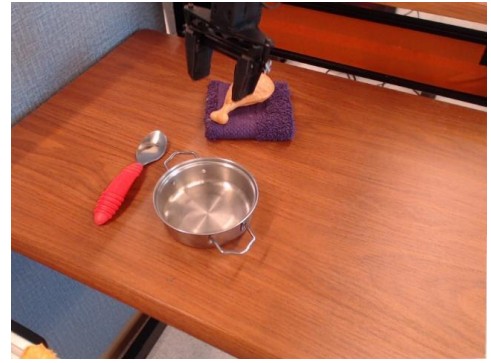

**Example of Bridgevqa Data**

Question: Who is positioned more to the left, the red spoon or the robot arm?

Answer: Red spoon is more to the left.

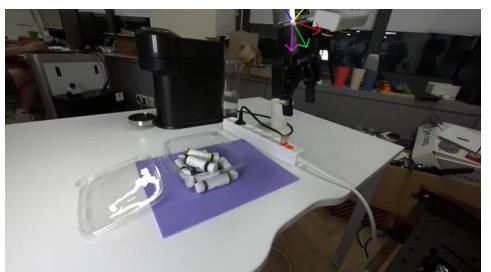

**Example of Robo2VLM Data**

Question: The robot task is to flip the switch of the extension cord. Which colored arrow correctly shows the direction the robot will move next? Choices: A. Yellow. B. Blue. C. None of the above. D. Purple. E. Green. Please answer directly with only the letter of the correct option and nothing else.

Answer: C

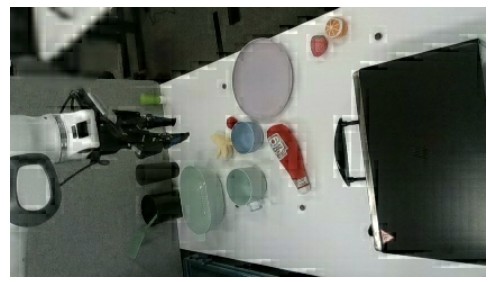

**Example of Robobrain Data (ShareRobot)**

Question: reach for the ketchup bottle

Answer: [[113.66,106.19],[153.30,111.42], [217.61,121.14],[248.27,135.35]]

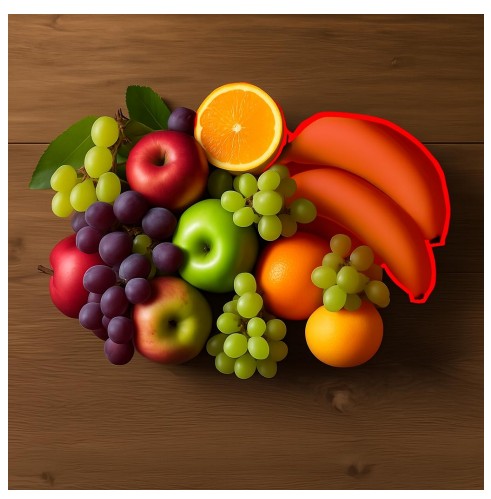

**Example of Omni-Generation Tasks**

Question: Use a red mask to segment the edges of the banana in the image.

Answer: Generated mask in the image.

## B  USAGE OF LLMS

In the final stages of preparing this manuscript, the authors used a Large Language Model (LLM) solely for grammar checking and language polishing. The model assisted in improving sentence structure and correcting grammatical errors to enhance readability.

