# OpenReview forum: "VLM4VLA: Revisiting Vision-Language-Models in Vision-Language-Action Models"
_ICLR.cc/2026/Conference — ICLR 2026 Poster_

### Official Review · Reviewer_ZXC9 · 2025-10-20

**Soundness:** 4
**Presentation:** 3
**Contribution:** 3
**Rating:** 6
**Confidence:** 4

**Summary:**

This work conducts an empirical study (in simulation) to investigate the relationship between the capabilities of the VLM backbone and its downstream performance in a VLA. The authors propose VLM4VLA, an adaptation pipeline that adds a MLP head to the VLM backbone to ensure a fair comparison across models. Through experiments in simulation, the authors arrive at three findings: 1) a VLM's performance on general VQA benchmarks is often a poor predictor of performance on robotic manipulation tasks, 2) fine-tuning a VLM on auxiliary embodied VQA tasks doesn't reliably improve performance, and 3) fine-tuning the VLM's vision encoder is necessary.

**Strengths:**

Large-scale study: the primary strength of this work is the sheer number of models (7) and simulation environments (3) studied. This comprehensive evaluation provides a valuable perspective

VLM4VLA Framework: by using an MLP (non-stochastic), the authors maintain a simple adaptation module which enables isolating the VLM backbone as one of the only independent variables. Ensuring prompts are adjusted per model is further evidence of clean experimentation.

Findings: the findings that VQA performance is not predictive of manipulation performance, in simulation, is good to know for future work. However, I would like to add that I think the main reason researchers leverage a VLM backbone is to capitalize on the potential of generalization, not in-distribution robot manipulation performance (this must be taught via imitation learning).

**Weaknesses:**

Reliance on simulation: the paper's most significant weakness, which the authors acknowledge, is exclusive use of simulated environments. While this choice was made for reproducibility and scale, it limits whether or not the paper's main claims generalize to the real-world, which is ultimately what is needed in robotics. The sim-to-real gap is a well-known challenge in robotics and care must be taken to extrapolate findings from simulation to the real world. Moreover, Figure 3 supports this statement: the findings of VLA performance and VLM capability are not consistent across simulation environments.

As another example of the potential pitfalls of simulation, as vision encoders are often pre-trained on real-world images, evaluating them purely in the simulated domain may add a confounding variable to the analysis, as most VLAs are only fine-tuned on real-world data. Therefore, I suggest that all main claims in the paper add the qualifier that they hold only in the simulation benchmarks tested.

Novelty of vision encoder finding: while the result that fine-tuning the vision encoder is crucial, this observation is not entirely new. Other works such as OpenVLA [1] and Otter [2] have described similar findings.

[1] Kim, M. J., Pertsch, K., Karamcheti, S., Xiao, T., Balakrishna, A., Nair, S., ... & Finn, C. (2024). Openvla: An open-source vision-language-action model. arXiv preprint arXiv:2406.09246.

[2] Huang, H., Liu, F., Fu, L., Wu, T., Mukadam, M., Malik, J., ... & Abbeel, P. (2025). Otter: A vision-language-action model with text-aware visual feature extraction. arXiv preprint arXiv:2503.03734.

**Questions:**

Clarification of Pi0 baseline: could you elaborate a bit more on the specifics of why the official implementation of Pi0 was not used? In particular, were the official weights of the Physical Intelligence model used?

Defining "VLM General Capability:" Figure 3 aggregates 'vlm capability' on the x-axis, but it is not precisely defined. What exactly does this mean?

---

> ### Author Response · Authors · 2025-11-23
> **Rebuttal to Review ZXC9 (Part1)**
>
> We sincerely thank you for your time and constructive feedback. We appreciate the insightful questions, which have helped us significantly strengthen our experimental analysis and clarify our contributions. In response to your suggestions, we have added a detailed discussion and conducted new experiments, which are summarized below.
>
> ---
>
> **1: I would like to add that I think the main reason researchers leverage a VLM backbone is to capitalize on the potential of generalization, not in-distribution robot manipulation performance (this must be taught via imitation learning) (S1)**
>
> We fully agree with your perspective! The primary motivation for integrating VLM backbones is indeed to leverage their innate semantic priors for generalization.
>
> This philosophy directly guided our choice of benchmarks. We selected CALVIN (ABC → D) specifically because it forces the model to generalize to unseen scenes (different textures, background colors, and object arrangements). Similarly, on our Simpler-Bridge experiments, training uses real-world BridgeV2 data, whereas testing is carried out in simulated, rendered environments; thus, the evaluation tests the model's ability to generalize to rendered images and actions, various instructions, and positions.
> The aforementioned forms of generalization are, of course,  insufficient to capture the full purpose of using a VLM as the backbone. We believe this is a key direction that warrants further investigation in future work.
>
>
>
> ---
>
> **2: About reliance and potential pitfalls for vision encoder of simulation (W1, W2)**
>
> We fully accept the reviewer's suggestion. We agree that the simulation-based nature of the experiments is a limitation, and we will explicitly add the qualifier to our claims in the revised manuscript.
>
> #### **(1) Clarification on Simulation Reliance & Generalization**
> - *Justification for Simulation*
>
> As discussed in Sec. 5, our priority for this study was to rigorously compare the architectural differences between VLM backbones. Real-robot experiments introduce uncontrollable variance (e.g., lighting changes, reset noise) that often obscures subtle performance differences between models. However, even a 5% difference in success rate could lead to qualitatively different conclusions. Ensuring that every real-robot trial is exactly identical across runs (e.g., initial arm configuration, object placement, lighting, distractors) and controlled identically across models is extremely challenging. Simulation provides a deterministic environment, ensuring that every trial is identical across baselines. This maximizes internal validity, allowing us to isolate the specific contributions of the model architecture.
> - *Mitigating the Gap (SIMPLER)*
>
> To address the generalization concern, in our paper, we utilized the SIMPLER benchmark [1]. This suite minimizes the sim-to-real gap by training on real-world data (BridgeV2) and evaluating in a simulator that uses visual matching techniques (e.g., green screening, texture matching) to align with the real world. [1] has demonstrated a positive correlation between SIMPLER success rates and real-world execution, suggesting our findings are indicative of real-world potential. Furthermore, since VLA tasks rely primarily on images and texts rather than complex contact dynamics, the "sim-to-real" gap is primarily visual, which SIMPLER is designed to bridge.
>
> #### **(2) Addressing the *Confounding Variable*: Is Fine-tuning only for the Visual Domain Gap?**
>
> The critical question raised is: Does the vision encoder require fine-tuning merely to bridge the visual gap (Real → Sim), or is there a mismatch between VLM understanding features and robot control features?
>
> To disentangle these two factors, we conducted a controlled experiment using real-world data. If the need for fine-tuning were solely due to the simulated domain visual gap, then fine-tuning on real-world images (which align with the VLM's pre-training) should not require updating the vision encoder.
>
> We utilized the BridgeV2 dataset, which consists entirely of real-world images. Following the procedure in *Sec. 4.2*, we introduced a new VLM fine-tuning task to inject the control-related information to the VLM backbone on real-world images. We encoded BridgeV2 actions into discrete tokens using Fast-Token [2] and fine-tuned Qwen3VL-4B to predict these action tokens autoregressively (formatting it as a QA task). We compare three settings of finetuning the VLM:
> - Baseline: without finetuning the VLM at all
> - Freeze Vision FT: Fine-tuning only the LLM (keeping the vision encoder frozen).
> - Unfreeze Vision FT: Fine-tuning both the LLM and the vision encoder.

---

> > ### Author Response · Authors · 2025-11-23
> > **Rebuttal to Review ZXC9 (Part2)**
> >
> > (Continuing from the previous section)
> >
> > We then took these modified VLM backbones as initialization and trained a standard VLM4VLA policy with frozen vision encoder on Simpler-Bridge.
> >
> > | Qwen3VL-4B                                                   | Simpler-Bridge |
> > | ------------------------------------------------------------ | -------------- |
> > | Baseline + freeze VLA vision                            | 27.6           |
> > | **Freeze Vision FT** + freeze VLA vision       | 28.0 *(+0.4)* |
> > | **Unfreeze Vision FT** + freeze VLA vision | 45.7 *(+18.1)*  |
> >
> > The results reveal a critical insight. Even when training on real-world images (where there is no sim-to-real visual gap), keeping the vision encoder frozen (Row 2) fails to improve downstream performance. The VLM's native visual features—despite being "real-world" are insufficient for control. When we allow the vision encoder to update on real-world control data (Row 3), performance improves dramatically (+18.1%).
> >
> > This demonstrates that the necessity of fine-tuning the vision encoder is not primarily a confounding variable caused by simulation artifacts. Instead, we consider it be driven by a semantic gap: the visual features optimized for multimodal understanding or reasoning are not inherently aligned with the fine-grained representations required for low-level manipulation, regardless of whether the images are real or simulated.
> >
> > [1] Li, X., Hsu, K., Gu, J., Pertsch, K., Mees, O., Walke, H. R., ... & Xiao, T. (2024). Evaluating real-world robot manipulation policies in simulation. arXiv preprint arXiv:2405.05941.
> >
> > [2] Pertsch K, Stachowicz K, Ichter B, et al. Fast: Efficient action tokenization for vision-language-action models[J]. arXiv preprint arXiv:2501.09747, 2025.
> >
> > ---
> >
> > **3: Novelty of vision encoder finding: while the result that fine-tuning the vision encoder is crucial, this observation is not entirely new. Other works such as OpenVLA [1] and Otter [2] have described similar findings.**
> >
> > Thank you for pointing this out. We have already cited OpenVLA[1] in the paper, and we will add citations and a brief description for Otter[2] as well.
> >
> > [1] Kim, M. J., Pertsch, K., Karamcheti, S., Xiao, T., Balakrishna, A., Nair, S., ... & Finn, C. (2024). Openvla: An open-source vision-language-action model. arXiv preprint arXiv:2406.09246.
> >
> > [2] Huang, H., Liu, F., Fu, L., Wu, T., Mukadam, M., Malik, J., ... & Abbeel, P. (2025). Otter: A vision-language-action model with text-aware visual feature extraction. arXiv preprint arXiv:2503.03734.
> >
> > ---
> >
> > **4: Clarification of Pi0 baseline (Q1)**
> >
> > The primary reason for not using the official Pi0 checkpoint was to control for pre-training data.
> >
> > To ensure a fair comparison between the Pi0 approach and other VLMs, it was crucial to standardize the starting point. The official Pi0 checkpoints were trained on Physical Intelligence’s massive proprietary dataset. Instead, we integrated the Pi0 architecture into our framework but initialized the backbone with standard pre-trained VLM weights (identical to the other baselines). Under this setup, all models are trained and evaluated on exactly the same datasets. This allows us to assess the effectiveness of the Pi0 architecture itself versus other VLA designs, independent of the advantage provided by private, large-scale pre-training data. Further implementation details are available in *Appendix A.3*.
> >
> > ---
> >
> > **5: Clarification of details in Figure 3 (Q2)**
> >
> > In Figure 3, we use established multimodal understanding benchmarks (various VQA suites) to assess the general performance of VLMs. Concretely, InternVL and QwenVL are general-purpose VLMs that have been evaluated across a wide range of multimodal understanding benchmarks. For Fig3, we take the average of their officially reported results on a set of multimodal general, reasoning, text, grounding, agentic, and embodied benchmarks as the x-axis value. This set includes: MMBench v1.1 (en), MMStar, BLINK, HallusionBench, AI2D, OCRBench, MVBench, VideoMME, MMMU, MathVista, MathVision, CountBenchQA, RefCOCO (all), GPQA, OmniSpatial, RealWorldQA, ERQA, and VSI-Bench.
> > For PaliGemma and Kosmos, which are not trained for general-purpose tasks, we use their performance on their respective specialized evaluation suites as the metric. Specifically, PaliGemma is evaluated on RefCOCO (all), AI2D, CountBenchQA, and GQA; Kosmos is a grounding-focused VLM, which use only RefCOCO results as its metric.
> > Because the latter two models are evaluated on a subset of the tasks used for the former two (reflecting their lack of general-task capability), we apply a downweighting to their scores. This yields an approximate assessment of the “general capability of different types of VLMs” as a reference.
> >
> > ---
> >
> > Thank you again for your time and effort in reviewing our work! We hope this clarification can solve your concerns!

---

### Official Review · Reviewer_KoNy · 2025-10-29

**Soundness:** 3
**Presentation:** 3
**Contribution:** 4
**Rating:** 8
**Confidence:** 4

**Summary:**

The authors propose VLM4VLA, a general framework for adapting arbitrary Vision-Language-Models (VLMs) into Vision-Language-Action (VLA) models, requiring only a minimal number of new parameters. The work also investigates how the choice of the base VLM affects the downstream VLA's performance.

**Strengths:**

1. The paper presents a framework for fairly comparing the performance of different VLMs on VLA tasks and provides an in-depth study into the reasons for performance discrepancies.
2. By using an MLP action head instead of a more complex diffusion-based one, the framework avoids introducing stochasticity. This ensures a "fair and reproducible" comparison across the different VLMs.
3. It systematically proposes three benchmarks for evaluating VLM capabilities: general capability, embodied-specific capability, and the vision encoder.
4. The experiments cover a wide range of models and test tasks, providing a strong empirical baseline for future work.

**Weaknesses:**

1. The study lacks real-robot experiments. The sim-to-real gap is a major concern in the VLA field, and this work doesn't clarify how different VLMs might affect the model's final generalization to real-world scenarios.
2. Diffusion action heads and MLP action heads may leverage VLM capabilities differently (e.g., many diffusion heads use the VLM's KV-cache for information interaction). The paper does not directly compare the impact of these two approaches on VLA performance.

**Questions:**

1. Could you provide a performance comparison and analysis for a model using both a diffusion action head and an MLP action head? Or, perhaps more directly, a comparison between using embeddings vs. using the KV-cache for information interaction?
2. Are there any additional experiments that could demonstrate the difference in generalization capabilities among the various VLMs when applied to VLA tasks?

---

> ### Author Response · Authors · 2025-11-23
> **Rebuttal to Review KoNy (Part1)**
>
> We sincerely thank you for your time and constructive feedback. We appreciate the insightful questions, which have helped us significantly strengthen our experimental analysis and clarify our contributions. In response to your suggestions, we have added a detailed discussion and conducted new experiments, which are summarized below.
>
> ---
>
> **1: The sim-to-real gap is a major concern in the VLA field, and this work doesn't clarify how different VLMs might affect the model's final generalization to real-world scenarios. (W1)**
>
> Thanks for your insightful comments! We fully agree that real-world validation is the ultimate goal for VLA research. However, as discussed in Sec. 5, our priority was to rigorously compare the architectural differences between VLM backbones for this study. Real-robot experiments often introduce uncontrollable variance (e.g., lighting changes, reset noise) that can obscure subtle performance differences between models. However, even a 5% difference in success rate could lead to qualitatively different conclusions. Ensuring that every real-robot trial is exactly identical across runs (e.g., initial arm configuration, object placement, lighting, distractors) and controlled identically across models is extremely challenging. Simulation provides a reproducible environment, ensuring that every trial is identical across baselines so that our results reflect only the model's capability.
>
> To mitigate the gap, we evaluate and report the performance on multiple simulators to reduce the simulation-specific bias. What's more, since VLA tasks rely primarily on visual-semantic understanding (images, text, and action) rather than complex contact dynamics (e.g., detailed friction modeling as in locomotion), we consider that performance on high-fidelity benchmarks like SIMPLER and Calvin is a good indicator of real-world potential. Especially for benchmarks like SIMPLER is explicitly designed to correlate with real-world performance by bridging the visual gap (using real textures and backgrounds) [1].
>
> In the end, we fully agree that real-world deployment is very important and we have highlighted this limitation in our conclusion section. We are actively working on transferring these findings to physical platforms for future studies.
>
> [1] Li, X., Hsu, K., Gu, J., Pertsch, K., Mees, O., Walke, H. R., ... & Xiao, T. (2024). Evaluating real-world robot manipulation policies in simulation. arXiv preprint arXiv:2405.05941.
>
>
> ---
>
> **2: About diffusion action heads and MLP action heads (W2,Q1)**
>
> These two types of action heads can indeed yield different VLA performance. Here we present results on Simpler-Bridge using the Qwen3VL-4B model with two action heads—an MLP head (as in VLM4VLA) and a diffusion head (flow-matching) that leverages the KV-cache—evaluated across multiple rollouts of multiple checkpoints at different training steps. We report both the mean and variance.
>
> |                | Qwen3VL-4B+MLP （VLM4VLA） | Qwen3VL-4B+Diffusion |
> | -------------- | -------------------------- | -------------------- |
> | Simpler-Bridge | 52.3$\pm$4.2                   | 47.8$\pm$12.1            |
>
> We observed that flow-matching or diffusion-based methods introduce stochasticity into the policy: results vary substantially both across checkpoints at different training steps and across multiple rollouts from the same checkpoint. To keep the VLA architecture simple and consistent, we therefore adopt a lightweight MLP action head.
>
> We further clarify that our VLA architecture (MLP action head) actually leverages the same KV-cache information used by diffusion action heads, but in a more lightweight “action query” manner rather than via an “action expert” module:
> - We append a learnable action query token to the end of the VLM input. During the forward pass, this token attends to the KV features of other VLM tokens, updating its own representation. In this sense, the action query token fully exploits the VLM’s internal KV-cache.
> - For action decoding, we take the action query feature—which has incorporated KV-cache signals (including final-layer embeddings)—as input to the MLP head, which outputs the action.
>
> Thus, our approach is conceptually similar to diffusion action head methods such as Pi0: both use deep internal VLM representations (KV-cache) as features for action decoding. In Tables 1 and 2 of the paper, comparing PaliGemma-1 in VLM4VLA with Pi0 (also uses Paligemma-1 as backbone) shows very similar performance across all three benchmarks, further supporting the above conclusion.
> For more details about VLA architecture design, please refer to *Sec. 3.2* and *Appendix A.3*.

---

> > ### Author Response · Authors · 2025-11-23
> > **Rebuttal to Review KoNy (Part2)**
> >
> > ---
> >
> > **3: Are there any additional experiments that could demonstrate the difference in generalization capabilities among the various VLMs when applied to VLA tasks? (Q2)**
> >
> > Thanks for this question! We are also actively exploring more generalization tasks. The existing benchmarks in VLM4VLA have evaluated certain aspects of VLA generalization. In CALVIN, training is conducted on the ABC scenes, while testing is performed on the D scene—requiring the model to handle changes in object colors, positions, background colors, and language instructions.
> > In Simpler-Bridge, training uses real-world BridgeV2 data, whereas testing is carried out in simulated, rendered environments; thus, the evaluation requires generalization to rendered images and actions, different language instructions, initial robot poses, and target object locations. We also note that recent benchmarks have focused on other critical dimensions of generalization. For instance, INT-ACT [1] and Align [2] introduce out-of-distribution objects in simulation to probe semantic robustness, while [3] employs emoji cards to assess whether VLA policies can correctly ground abstract concepts. In future work, we intend to expand our evaluation suite to encompass this broader range of generalization capabilities.
> >
> >
> > [1] Fang I, Zhang J, Tong S, et al. From intention to execution: Probing the generalization boundaries of vision-language-action models[J]. arXiv preprint arXiv:2506.09930, 2025.
> >
> > [2] Kachaev N, Kolosov M, Zelezetsky D, et al. Don't Blind Your VLA: Aligning Visual Representations for OOD Generalization[J]. arXiv preprint arXiv:2510.25616, 2025.
> >
> > [3] Zhang, C., Yang, R., Chen, X., Wang, K., Zhao, L., Chen, Y., & Bian, J. (2025). How Do VLAs Effectively Inherit from VLMs?. arXiv preprint arXiv:2511.06619.
> >
> > ---
> >
> > Thank you again for your time and effort in reviewing our work! We hope our clarification can solve all your concerns and shows the improved quality of our paper!

---

> ### Comment · Reviewer_KoNy · 2025-11-26
>
> The authors have addressed my concerns, and I have decided to maintain my rating.

---

> > ### Author Response · Authors · 2025-11-28
> >
> > We are happy to hear that our response addressed your concerns. If you have any further questions or suggestions, please do not hesitate to let us know.

---

### Official Review · Reviewer_n8E9 · 2025-10-31

**Soundness:** 3
**Presentation:** 3
**Contribution:** 3
**Rating:** 8
**Confidence:** 4

**Summary:**

This paper studies the impact of the underlying VLM on VLA policy performance through the VLM4VLA architecture. VLM4VLA trains a continuous regression head on top of the VLM for the robot action prediction task. The paper trains and evaluates VLAs based on several different VLM sizes across the CALVIN, SimplerEnv, and Libero simulated environments. The paper finds that the impact of base VLM capabilities varies greatly by environment. Furthermore, continuing to train the VLM on embodiment and 3D-specific tasks produces a slight degradation in VLA performance.

**Strengths:**

1. The paper addresses the important problem of the impact of VLMs on VLA performance, which has been relatively understudied in prior work.
1. The paper shows that general VLM capability does not necessarily correlate with VLA capability. This is an important finding since it contradicts the common intuition that a stronger VLM model is always better for VLAs. For example, recent VLA works use newer VLM bases, which this study shows is not necessarily a good decision.
1. The paper supports its claims with comprehensive empirical analysis across many different base VLMs and simulated environments.
1. The result showing that further training on VLM auxiliary tasks does not improve downstream VLA performance, as presented in Figure 4, is a novel insight. The paper supports this claim with comprehensive results that test a wide variety of auxiliary tasks. Like the VLM-to-VLA transfer result, this finding is also surprising and contrary to prior efforts that design auxiliary tasks to improve VLM embodied capabilities.
1. The results in Section 4.3 provide an important lesson by comprehensively demonstrating the importance of training the visual encoder, where freezing the visual encoder leads to a large performance drop.
1. The VLM4VLA architecture provides a consistent and simple way to adapt VLMs to VLAs. The paper also shows that this architecture produces results superior to prior works that use more complicated designs.
1. The paper includes sufficient reproduction details in the appendix.

**Weaknesses:**

1. The importance of the visual encoder can also be explained by several other factors beyond the need to finetune it. (1) The Qwen2.5-VL model is sensitive to image resolution, with higher resolutions using more visual tokens per image and typically producing better performance. It is possible that the visual encoder could be frozen if the image resolution were increased. (2) The VLMs are trained primarily on real images, while the selected benchmarks are not photorealistic and use only simple rendering (see the top of Figure 1). Thus, the visual encoder may need to be finetuned to overcome this domain gap, which would likely not be an issue with real robot data.
2. The linear correlation plot in Figure 3 is hard to interpret (see Question 2). The paper should report the strength of the correlation.
3. The paper appears to rely on VQA benchmarks as a measure of VLM capability in Figure 3. Are there more specific VLM capabilities, such as spatial understanding, that have a direct relationship with VLA performance?
4. The paper does not provide analysis of what drives VLA performance based on the base VLM. For example, why does Kosmos-2 achieve the highest success rate in Table 2? While showing that VLM capabilities do not necessarily correspond to VLA capabilities is a valuable contribution, providing initial evidence for why this occurs would strengthen the paper.
5. The paper does not experiment with larger VLM sizes. It is possible that larger VLMs learn more general features that improve VLA capabilities, which this paper does not rule out.

**Questions:**

1. Is it still important to train the visual encoder if the image resolution is increased, as discussed in Section 4.3 (see Weakness 1)?
2. What do the colors of the lines and shapes represent in Figure 3? Likewise, which points are used to fit the lines?
3. How does the paper compare the VLM performance of the models between “proprietary tasks” and “general-purpose VQA benchmarks” (L357–368)? Shouldn’t VLM general capability be assessed using the same benchmarks for all models?

---

> ### Author Response · Authors · 2025-11-23
> **Rebuttal to Review n8E9 (Part1)**
>
> We sincerely thank you for your time and constructive feedback. We appreciate the insightful questions, which have helped us significantly strengthen our experimental analysis and clarify our contributions. In response to your suggestions, we have added a detailed discussion and conducted new experiments, which are summarized below.
>
> ---
>
> **1: About the importance of the vision encoder (W1, Q1)**
>
> In Section 4.3, we demonstrate that freezing the vision encoder leads to a significant performance drop. Thank you for proposing alternative explanations for this phenomenon! We have investigated both hypotheses below.
>
> #### **(1) Image Resolution (W1, Q1)**
>
> We appreciate this insightful hypothesis. We are also curious about the influence of that image resolution on the final performance. To explore the effectiveness, we conducted higher-resolution experiments on Qwen2.5-VL-3B and Qwen2.5-VL-7B. Specifically, during VLA training, we set the input image resolution to 224, 512, and 768, respectively, and evaluated on the Calvin ABC-D task under both settings: freezing and not freezing the vision encoder. The results are as follows.
>
> |                         | 224x224          | 512x512          | 768x768          |
> | ----------------------- | ---------------- | ---------------- | ---------------- |
> | Qwen2.5VL-3B            | 3.856            | 3.832            | 3.836            |
> | + freeze vision encoder | 2.855 *(-1.001)* | 2.749 *(-1.083)* | 2.715 *(-1.121)* |
> | Qwen2.5VL-7B            | 4.057            | 4.062            | 3.998            |
> | + freeze vision encoder | 2.823 *(-1.234)* | 2.746 *(-1.316)* | 2.693 *(-1.305)* |
>
>
> As shown in the table, simply increasing the input resolution (from 224 to 768) did not significantly improve VLA performance when the encoder is fine-tuned. These results suggest that the necessity of fine-tuning is not merely a compensation for low resolution.
> However, the performance degradation caused by freezing the encoder became more pronounced at higher resolutions (e.g., the gap widens from -1.00 to -1.12 for the 3B model). A plausible explanation is that higher resolution and more frozen visual tokens may lead to spurious correlation which incurring additional performance losses.
>
> #### **(2) Real to Sim Visual Domain Gap of  Vision Encoder (W1)**
>
>
> To address this, we refer to our results on the Simpler-Bridge benchmark (Section 4.3). Crucially, Simpler-Bridge uses the **real-world BridgeV2 dataset** for training. These training images are collected in real-world robot platform. As for the simulator, Simpler uses techniques like green screening and texture matching to reduce the visual appearance gap between real environments and raw simulation [1]. Despite this alignment in visual domains, we still observed that freezing the vision encoder degrades performance compared to fine-tuning. This result indicates that the visual encoder requires adaptation not just to handle domain gap in simulation artifacts, but to **learn control-specific information (like low-level actions) that are not captured during general web-scale pre-training**.
>
> To further evaluate the "Confounding Variable" hypothesis, we add a new experiment to address this question: Does the vision encoder require fine-tuning merely to bridge the visual gap (Real → Sim), or is there a mismatch between VLM understanding features and robot control features?
>
> To disentangle these two factors, we conducted a controlled experiment using real-world data. If the need for fine-tuning were solely due to the simulated domain visual gap, then fine-tuning on real-world images (which align with the VLM's pre-training) should not require updating the vision encoder.
>
> We utilized the BridgeV2 dataset, which consists entirely of real-world images. Following the procedure in *Sec. 4.2*, we introduced a new VLM fine-tuning task to inject the control-related information to the VLM backbone on real-world images. We encoded BridgeV2 actions into discrete tokens using Fast-Token [2] and fine-tuned Qwen3VL-4B to predict these action tokens autoregressively (formatting it as a QA task). We compare three settings of VLM finetuning:
> - Baseline: without finetuning the VLM at all
> - Freeze Vision FT: Fine-tuning only the LLM (keeping the vision encoder frozen).
> - Unfreeze Vision FT: Fine-tuning both the LLM and the vision encoder.
>
> We then took these modified VLM backbones and trained a standard VLM4VLA policy with frozen vision encoder on Simpler-Bridge.
>
>
> | Qwen3VL-4B                                                   | Simpler-Bridge |
> | ------------------------------------------------------------ | -------------- |
> | Baseline + freeze VLA vision                            | 27.6           |
> | **Freeze Vision FT** + freeze VLA vision        | 28.0 *(+0.4)* |
> | **Unfreeze Vision FT** + freeze VLA vision| 45.7 *(+18.1)*  |

---

> ### Author Response · Authors · 2025-11-23
> **Rebuttal to Review n8E9 (Part2)**
>
> (Continuing from the previous section)
>
> The results reveal a critical insight. Even when training on real-world images (where there is no sim-to-real visual gap), keeping the vision encoder frozen (Row 2) fails to improve downstream performance. The VLM's native visual features—despite being "real-world" are insufficient for control. When we allow the vision encoder to update on real-world control data (Row 3), performance improves dramatically (+18.1%).
>
> This demonstrates that the necessity of fine-tuning the vision encoder is not primarily a confounding variable caused by simulation artifacts. Instead, we consider it be driven by a semantic gap: the visual features optimized for multimodal understanding or reasoning are not inherently aligned with the fine-grained representations required for low-level manipulation, regardless of whether the images are real or simulated.
>
>
>
> [1] Li, X., Hsu, K., Gu, J., Pertsch, K., Mees, O., Walke, H. R., ... & Xiao, T. (2024). Evaluating real-world robot manipulation policies in simulation. arXiv preprint arXiv:2405.05941.
>
> [2] Pertsch K, Stachowicz K, Ichter B, et al. Fast: Efficient action tokenization for vision-language-action models[J]. arXiv preprint arXiv:2501.09747, 2025.
>
>
>
> ---
>
> **2: About More Details about Figure 3 (W2, Q2)**
>
> Thanks for your insightful comments. We will revise the manuscript, and we summarize the explanation of Figure 3 as below:
> - The x-axis represents the general capability of different VLMs.
> - Each set of three vertically aligned points with the same color corresponds to the results of the same VLM used as the VLM4VLA backbone across three benchmarks.
> - Marker shapes denote benchmarks: circles for SimplerEnv, triangles for LIBERO, and squares for CALVIN (see the legend on the right).
> - The three colored lines are linear fits to the points from each benchmark: red for Simpler (fit over circle markers), blue for LIBERO (fit over triangle markers), and green for CALVIN (fit over square markers).
>
> Figure 3 uses seven VLMs (with data sourced from the experiments reported in Tables 1 and 2). To quantify the strength of the linear relationship between each benchmark and VLM capability, we report the correlation metrics for the three fitted lines (Pearson correlation coefficient $r$ and $R^2$):
> - Calvin: $r=0.919, R^2=0.844$
> - SimplerEnv: $r=-0.321, R^2=0.103$
> - Libero: $r=0.381, R^2=0.145$
>
> ---
>
> **3: Comparing general VLM capabilities across different benchmarks and more specific VLM capabilities that have a direct relationship with VLA performance. (W3, Q3)**
>
> Thank you for this insightful question. We address the relationship between specific capabilities and VLA performance in two ways:
>
> (1) Clarifying the *VQA* Metric (Fig. 3): We would like to clarify that the metric labeled *VQA* in Figure 3 is not limited to simple Question-Answering. It is a composite score representing a broad spectrum of multimodal capabilities, including explicit spatial and grounding tasks. Specifically, the score aggregates performance across domains such as:
> - General Multi-modal Understanding: MMBench v1.1 (en), MMStar, VideoMME, MMMU
> - Spatial & Embodied Understanding: BLINK, ERQA, RealWorldQA, OmniSpatial, MVBench, VSI-Bench
> - Grounding: RefCOCO, OCRBench, CountBenchQA
> - Reasoning & Agentic: MathVista, MathVision, AI2D, HallusionBench, GPQA.
>
>
> For general-purpose models (InternVL, QwenVL), we averaged results across this full suite. For specialized models (PaliGemma, Kosmos), we utilized their relevant subsets (e.g., RefCOCO for Kosmos) to construct a representative "General Capability" score. Thus, the correlation shown in Fig. 3 already inherently captures spatial understanding and grounding capabilities.
>
> (2) Investigating Specific Capabilities: Beyond observing correlations, we performed controlled interventions in Sec. 4.2 to verify if enhancing specific capabilities improves VLA performance. We explicitly fine-tuned the Qwen2.5-VL backbone on specialized tasks, including Pointing (RoboPoint), Spatial Understanding (Robo2VLM, VICA-332k), Embodied Understanding (BridgeVQA, RoboBrain2), and T2I generation tasks.
>
> Surprisingly, our results indicate that maximizing these specific metrics via fine-tuning does not yield significant gains in robotic manipulation. In fact, we observed that fine-tuning on these narrow domains often degrades the model's general representational power (catastrophic forgetting), leading to worse VLA performance. For more information and details, please refer to Sec. 4.2 and Appendix A.5.

---

> > ### Author Response · Authors · 2025-11-23
> > **Rebuttal to Review n8E9 (Part3)**
> >
> > ---
> >
> > **4: Analysis of what drives VLA performance based on the base VLM (W4)**
> >
> >
> > We appreciate this thoughtful comment. This observation that static benchmarks (like VQA) do not perfectly predict dynamic robotic performance is one of the central findings of our empirical study.
> >
> > We believe the reason behind this observation is still an open question. Our hypothesis is that, VLM pre-training acts as a necessary condition (lower bound) but not a sufficient condition (upper bound).
> >
> > - Lower Bound: We conducted an ablation study to quantify the impact of VLM initialization (detailed below). Our results demonstrate that training from scratch (random initialization, without pretrained VLM) leads to substantial performance degradation. This confirms that the strong semantic representations acquired during pre-training are an prerequisite for effective policy learning in the VLA framework
> >
> > - Upper Bound: Once a model possesses sufficient semantic understanding, its success in dynamic manipulation depends on factors orthogonal to static VQA metrics—such as the architecture's adaptability to continuous control dynamics.
> >
> > This suggests that *robotic control* is a distinct capability dimension that is largely orthogonal to standard VQA (Multi-modal understanding) metrics once a certain level of semantic competency is reached. Therefore, we advise that while high static VLM scores are a good starting point for candidate selection, they cannot replace specific embodied evaluation, as the correlation breaks down at the top end of model performance.
> >
> > We conducted additional experiments training Qwen2.5-VL (3B & 7B) and PaliGemma from random initialization on both the Calvin ABC-D and Simpler-Bridge benchmarks. We strictly maintained the same training hyperparameters and evaluation protocols reported in the main paper.
> > As shown in the table below, models trained from scratch suffer from substantial performance degradation across all settings. This stark difference confirms VLM pre-training are fundamental to the generalization ability of the VLA model, rather than the architecture alone.
> >
> >
> > |              | Calvin ABC-D | Simpler-Bridge |
> > | ------------ | ------------ | -------------- |
> > | Qwen2.5VL-3B | 3.856        | 48.00          |
> > | + from scratch    |   1.381 *(-2.475)*           |    15.75 *(-32.25)*           |
> > | Qwen2.5VL-7B | 4.057        | 46.75          |
> > | + from scratch    |       1.769 *(-2.288)*      |      18.20 *(-28.75)*      |
> > | Paligemma-1  | 3.506       | 55.25          |
> > | + from scratch    |      1.129 *(-2.377)*       |    14.50 *(-40.75)*           |
> >
> >
> >
> > ---
> >
> > **5: About experiment with larger VLM sizes (W5)**
> >
> > Larger VLMs can indeed learn features with stronger generalization. However, due to resource constraints and inference-frequency requirements during policy deployment, prior VLA systems typically adopt VLM backbones no larger than 7B (often within 3B). Accordingly, our experiments focus on VLM sizes that cover the backbone range commonly used in VLA (0–10B).
> > In addition to the VLMs reported in the paper, we also evaluated Qwen3-VL variants with 2B, 4B, and 8B parameters using our framework. These new model evaluations do not change our conclusions: we still do not observe a clear correlation between general VLM capability and VLA performance, and larger models do not consistently yield better performance on VLA benchmarks. With additional resources, we plan to add results for a larger MoE model, 30A3B (a 30B-parameter mixture-of-experts), for investigation.
> >
> > ---
> >
> > Thank you again for your time and effort in reviewing our work! We hope our clarification can solve all your concerns, and we are always ready to answer any further questions!

---

### Official Review · Reviewer_C7oU · 2025-11-02

**Soundness:** 3
**Presentation:** 3
**Contribution:** 3
**Rating:** 6
**Confidence:** 3

**Summary:**

This paper studies the role of VLMs as backbones for VLA models. The authors fine-tune 7 open VLMs on robot simulation data and evaluate them across three benchmarks (Calvin, SimplerEnv, Libero). Their results show that a simple architecture and training objective can achieve competitive performance against recent models with specific VLMs. The authors also show that performance on Calvin is correlated with the VLM’s VQA performance, and that fine-tuning on VQA data from robot tasks does not help improve performance. Finally, the authors show the importance of fine-tuning the vision encoder of the VLM.

**Strengths:**

1. A meta-analysis of the role of VLMs in VLA models, showing competitive performance with a simplified training framework.
2. A study of the relationship between robot task performance and generic VQA performance.
3. A study of the (lack of) usefulness of VQA data extracted from robot data.
4. An analysis of the importance of fine-tuning the vision encoder, likely due to the domain mismatch between pretraining and robot data.

**Weaknesses:**

1. While the experiments show that some VLMs can achieve competitive performance in the VLM4VLA framework, the paper lacks specific guidelines and insights into why specific models perform better. While the authors found VQA performance to be predictive of Calvin performance, this was not the case for the other benchmarks. This left me wondering how I would choose the next VLM to initialize my VLA from.
2. I am also concerned about the decision of using the same hyperparameters for all the models. While this drastically lowers the number of experiments, models of different sizes would likely require different hyperparameters to perform best. This can affect all the results and conclusions that the authors derive from their experiments.

**Questions:**

1. Which VQA benchmarks were used for the correlation analysis in Fig 3?
2. Have you tried correlating robotic task performance with other downstream tasks besides VQA?
3. What is the (lower-bound) performance of a VLM4VLA initialized from scratch?

---

> ### Author Response · Authors · 2025-11-23
> **Rebuttal to Review C7oU (Part1)**
>
> We sincerely thank you for your time and constructive feedback. We appreciate the insightful questions, which have helped us significantly strengthen our experimental analysis and clarify our contributions. In response to your suggestions, we have added a detailed discussion and conducted new experiments, which are summarized below.
>
> ---
>
> **1: Insights and guidelines on current experiment results (W1)**
>
> We appreciate this thoughtful comment. This observation that static benchmarks (like VQA) do not perfectly predict dynamic robotic performance is one of the central findings of our empirical study.
>
> We believe the reason behind this observation is still an open question. Our hypothesis is that, VLM pre-training acts as a necessary condition (lower bound) but not a sufficient condition (upper bound).
>
> - Lower Bound: As shown in our "from-scratch" experiments in response 5, strong semantic representations from pre-training are essential to get off the ground.
>
> - Upper Bound: Once a model possesses sufficient semantic understanding, its success in dynamic manipulation depends on factors orthogonal to static VLM metrics—such as the architecture's adaptability to continuous control dynamics.
>
> This suggests that "robotic control" is a distinct capability dimension that is largely orthogonal to standard VLM metrics once a certain level of semantic competency is reached. Therefore, we advise that while high QA scores on various VLM benchmarks are a good starting point for candidate selection, they cannot replace specific embodied evaluation, as the correlation breaks down at the top end of model performance.
>
> ---
>
> **2: About choice of hyperparameters (W2)**
>
> Thank you for raising this important point regarding hyperparameter sensitivity across models.
> We would like to first explain our designs made in the paper to mitigate this difference, then we introduce a new experiment on hyperparameter sweep.
>
> For results reported in the paper, we have performed a sweep on $3$ different action chunk sizes during evaluation. We also use $5$ different checkpoints to mitigate the training instability. To be specific, for every result reported, we evaluated 5 distinct checkpoints (saved at different training steps) combined with 3 execution chunk settings, totaling 15 test runs per model. We reported the best performance among these runs. In most cases, the best performance was achieved before the final checkpoint, indicating that the models had already converged. This protocol ensures that our results reflect each model’s peak capability (convergence), effectively decoupling performance from training proficiency or randomness.
>
> To verify the robustness of our Learning Rate (LR) choice, we performed a sweep using values $\{1e-5,2e-5,5e-5,1e-4\}$ on representative models (Qwen2.5-VL-3B, Qwen2.5-VL-7B, and PaliGemma-1). As shown in the table below, while minor fluctuations exist, the downstream performance remains highly stable across this range, and the relative ranking of models remains consistent.
>
> | Learning Rate | 1e-5      | 2e-5      | 5e-5  | 1e-4  |
> | ------------- | --------- | --------- | ----- | ----- |
> | Qwen2.5VL-3B  | 3.844     | **3.856** | 3.832 | 3.792 |
> | Qwen2.5VL-7B  | 4.050     | **4.057** | 4.027 | 3.986 |
> | Paligemma-1   | **3.506** | **3.506** | 3.487 | 3.492 |
>
> Regarding batch size, we maximized this parameter uniformly across all models and make sure this stays the same. We believe that larger batch sizes benefit all architectures by stabilizing gradients; therefore, enforcing a large, identical batch size prevents any model from being disadvantaged by gradient noise.
> Further details on hyperparameter choices and the evaluation protocol can be found in *Appendices A.1 and A.2.1*.
>
>
> ---
>
> **3: Which VQA benchmarks were used for the correlation analysis in Fig 3? (Q1)**
>
> InternVL and QwenVL are general-purpose VLMs that have been evaluated across a wide range of multimodal understanding benchmarks. For Fig 3, we take the average of their officially reported results on a set of multimodal general, reasoning, text, grounding, agentic, and embodied benchmarks as the x-axis value. This set includes: MMBench v1.1 (en), MMStar, BLINK, HallusionBench, AI2D, OCRBench, MVBench, VideoMME, MMMU, MathVista, MathVision, CountBenchQA, RefCOCO (all), GPQA, OmniSpatial, RealWorldQA, ERQA, and VSI-Bench.
> For PaliGemma and Kosmos, which are not trained for general-purpose tasks, we use their performance on their respective specialized evaluation suites as the metric. Specifically, PaliGemma is evaluated on RefCOCO (all), AI2D, CountBenchQA, and GQA; Kosmos is a grounding-focused VLM, which use only RefCOCO results as its metric.
> Because the latter two models are evaluated on a subset of the tasks used for the former two (reflecting their lack of general-task capability), we apply a downweighting to their scores. This yields an approximate assessment of the “general capability of different types of VLMs” as a reference.

---

> > ### Author Response · Authors · 2025-11-23
> > **Rebuttal to Review C7oU (Part2)**
> >
> > ---
> >
> > **4: Have you tried correlating robotic task performance with other downstream tasks besides VQA? (Q2)**
> >
> > Thank you for this insightful question!
> > First, we would like to clarify that the "VQA" metric in our paper refers to the format of the evaluation (the model receives image and text inputs and produce answers) rather than a narrow task definition. As mentioned above, our VQA benchmark is actually a composite score representing a broad spectrum of visual–language capabilities. It aggregates performance across a variety of distinct downstream domains to evaluate the general capabilities of the VLM. Beyond broad correlations presented in Sec. 4.1.2 (Fig.3), we explicitly investigated how specific downstream tasks influence VLA performance in Sec. 4.2. To be specific, we fine-tune the same VLM on pointing (RoboPoint), spatial understanding (robo2vlm, VICA-332k), embodied understanding (RoboBrain2, BridgeVQA), and image generation tasks, and then use these finetuned models as VLA backbones for evaluation to test the correlation between robotic task and various downstream tasks.
> >
> >
> > ---
> > **5: What is the (lower-bound) performance of a VLM4VLA initialized from scratch? (Q3)**
> >
> >
> > Thank you for suggesting this baseline experiment. Establishing this lower bound is essential to quantify the specific contribution of VLM backbone.
> >
> > We conducted additional experiments training Qwen2.5-VL (3B & 7B) and PaliGemma from random initialization on both the Calvin ABC-D and Simpler-Bridge benchmarks. We strictly maintained the same training hyperparameters and evaluation protocols reported in the main paper.
> >
> > As shown in the table below, models trained from scratch suffer from substantial performance degradation across all settings. This stark difference confirms VLM pre-training are fundamental to the generalization ability of the VLA model, rather than the architecture alone.
> >
> >
> > |              | Calvin ABC-D | Simpler-Bridge |
> > | ------------ | ------------ | -------------- |
> > | Qwen2.5VL-3B | 3.856        | 48.00          |
> > | + from scratch    |   1.381 *(-2.475)*           |    15.75 *(-32.25)*           |
> > | Qwen2.5VL-7B | 4.057        | 46.75          |
> > | + from scratch    |       1.769 *(-2.288)*      |      18.20 *(-28.75)*      |
> > | Paligemma-1  | 3.506       | 55.25          |
> > | + from scratch    |      1.129 *(-2.377)*       |    14.50 *(-40.75)*           |
> >
> >
> > ---
> >
> > We hope our clarifications address your concerns and demonstrate the improved quality of our paper! Please feel free to reach out with any further questions.
> > Thank you again for your valuable time!

---

> ### Comment · Reviewer_C7oU · 2025-11-26
>
> Thank you for your responses and experiments supporting them. I have some follow-up questions, but I overall think the paper is in good shape and I will keep my positive recommendation.
>
> ---
>
> **1: Insights and guidelines on current experiment results (W1)**
>
> Thank you for confirming that VQA benchmarks do not well predict robotic performance, and that this is still an open question. I think it's important to stress this in the "Conclusion" of the paper. This does not reduce the great contributions of this work, but it will be an important and useful guidance for readers and future work in this direction.
>
> **2: About choice of hyperparameters (W2)**
>
> Thank you for highlighting the results. It might improve the paper if you could include a few words in the main body where you include Spearman rank correlation for different hyperparameters. This would act as a concise statistic for the results you have in Appendix, which would give the reader a notion of robustness of the results.
>
> **3: Which VQA benchmarks were used for the correlation analysis in Fig 3? (Q1)**
>
> Thanks for the clarification. Out of curiosity, have you tried measuring correlation across different sets of tasks that only include models that are capable of doing them? In other words, rather than downweighting their scores, you could simply not include models that do not support all sets of tasks, and instead report correlation on subsets of tasks of different sizes.
>
> **4: Have you tried correlating robotic task performance with other downstream tasks besides VQA? (Q2)**
>
> Thank you. Here, what I was trying to ask what level of correlation other VL tasks (e.g., image/video captioning, grounding via bbox prediction, image classification, etc) would have. Sorry for not being clear in my review. I don't expect an answer about this point so late in the rebuttal process.
>
> **5: What is the (lower-bound) performance of a VLM4VLA initialized from scratch? (Q3)**
>
> Thank you for running these experiments! I think it's useful to know what the lower-bound performance is.

---

> ### Author Response · Authors · 2025-11-28
>
> We sincerely thank you for your time and constructive feedback. We have updated our paper according to your advice and added some details to the paper according to the rebuttal.
>
> ---
>
> **Answer to 1: Insights and guidelines on current experiment results (W1)**
>
> Thank you for the suggestion. We have revised the conclusion section and emphasized this point.
>
> ---
>
> **Answer to 2: About choice of hyperparameters (W2)**
>
> We provide a brief overview of hyperparameter selection at the beginning of Sec. 3.3 and add detailed results and explanations in Appendix A.2. Thank you for the suggestion.
>
> ---
>
> **Answer to 3: Which VQA benchmarks were used for the correlation analysis in Fig 3? (Q1)**
>
> We are also very interested in the setting you described. In fact, we are actively expanding our pool of VLMs with general capabilities to ensure measuring corrlation across same sets of VLM benchmarks. If we remove PaliGemma and Kosmos from Table 3, only four models remain as data points, which reduces the reliability of linear fitting. To address this, we added two additional data points (Qwen3-VL-2B and Qwen3-VL-8B) for this comparison. We find that, even when restricting to general-purpose VLMs and evaluating across a unified set of benchmarks, general VLM capability still does not serve as a predictive indicator of VLA performance.
> We will include these new model data points, as well as additional running results on Qwen3VL-30A3B (an large 30B MoE model), in a subsequent version for the community’s reference.
>
> ---
>
> **Answer to 4: Have you tried correlating robotic task performance with other downstream tasks besides VQA? (Q2)**
>
> We apologize for the earlier misunderstanding.
> We present a correlation analysis between grounding capability (measured via bounding box prediction) and final VLA performance. We utilized the RefCOCO benchmark to quantify grounding accuracy, as this metric is applicable to all evaluated models, including PaliGemma and Kosmos. Specifically, we calculated the correlation between the RefCOCO scores of all seven models and their corresponding downstream VLA performance.
>
>
> The correlation coefficients for each benchmark are as follows: (Pearson correlation coefficient $r$ and $R^2$):
> - Calvin: $r=0.945, R^2=0.743$
> - SimplerEnv: $r=-0.492, R^2=0.150$
> - Libero: $r=0.671, R^2=0.276$.
>
> Based on these results, grounding also fails to serve as a reliable predictor of VLA performance. For the remaining VL tasks (captioning and classification), these foundational abilities are subsumed within the general-purpose VLM benchmarks; thus, we cannot straightforwardly isolate and evaluate models' ability on these VL tasks. We leave the investigation of the remaining VL tasks to future research.
>
> ---
>
> We are happy to hear your response. If you have any further questions or suggestions, please do not hesitate to let us know.

---

### Author Response · Authors · 2025-11-28

# Updated Manuscript

A revised version of our manuscript has now been uploaded.
We thank all Reviewers for their engagement that resulted in several of the modifications.

- We have revised the conclusion section and emphasized our findings in VLM4VLA.

- We provide a brief overview of hyperparameter selection at the beginning of Sec. 3.3 and add detailed results and explanations in Appendix A.2.

- We add a detailed explanation about the correlation analysis between VLM capability and VLA performance in Appendix A.4, further elaborating details in Figure 3.

- We add a discussion with related works in Sec 4.3.

- We add the training from scratch experiment to Sec 4.4.

---

### Author Response · Authors · 2025-12-01

Dear Area Chair and Reviewers,

We sincerely thank all reviewers for their thoughtful feedback. Given the recent AC reassignment, we would like to briefly summarize the status of our paper to support the AC’s final decision.

We have provided comprehensive responses to all questions raised by four reviewers.

- Reviewer KoNy assigned a score of **8** and  replied that all concerns has been addressed and would maintain the rating at **8**.
- Reviewer C7oU initially gave a score of **6**. After the first round of rebuttal, he raised several new clarifications, **noted that the paper is “in good shape,” and maintained a positive recommendation**. We designed new experiments addressing his concerns and submitted our responses. After our two rounds of discussion, the review thread was frozen.
- Reviewer n8E9 assigned a score of **8,** and Reviewer ZXC9 assigned a score of **6**, but neither has responded. We believe we have substantially addressed their concerns.

We believe all reviewers have read the paper carefully and offered multiple highly constructive suggestions. We conducted additional experiments and provided detailed discussions in the rebuttal. These efforts have improved the paper’s quality. Specifically, the main improvements we presented in the rebuttal are summarized as below.

**New Experiments, Enhancements and Insightful Conclusions**

- We conducted ablation studies of image resolution and training from scratch. (mentioned by Reviewer n8E9)
- We additionally include experiments based on diffusion action heads. (mentioned by Reviewer KoNy)
- We addressed the reviewer’s point about the importance of the vision encoder and designed additional fine-tuning experiments using Fast Tokens to further analyze the sources of the performance gap between VLMs and VLAs. Ultimately, we arrived at a **new insightful conclusion**: *the VLM–VLA gap arises not only from real-to-sim discrepancies, but also from a mismatch between the visual representations needed for language-understanding tasks and those required for low-level action control*. This new finding offers important guidance for future research in both VLA and VLM. (mentioned by Reviewer n8E9 and ZXC9)
- We have polished and updated the manuscript, added new experiment results, further clarifications to Figure 3 per the reviewers’ suggestions, incorporated the relevant citations, and emphasized the main contribution and insight in the Conclusion section. (mentioned by KoNy, C7oU, n8E9, ZXC9)

Sincerely,

The Authors

---

### Meta-Review · Area_Chair_7NoV · 2026-01-08

**Summary:**

Reviewers were broadly positive about the paper’s clean benchmarking setup and large empirical sweep, but they flagged a few recurring issues.
- The biggest concern was scope and external validity: because results are mostly from simulation benchmarks, reviewers wanted the claims clearly framed as “true for these benchmarks” and were cautious about how much this says about real-robot / sim-to-real behavior.

- Several reviewers also wanted more practical guidance—not just “VLM scores don’t predict VLA performance,” but some clearer intuition or evidence for why certain backbones work better and how a practitioner should choose a VLM.

- There were methodological worries about fairness (e.g., using shared hyperparameters across very different model sizes) and requests to make the Figure 3 correlation analysis easier to interpret and quantify.

- Finally, a couple reviewers noted that the vision-encoder fine-tuning takeaway may overlap with prior observations and asked whether it could be explained by confounds like image resolution or simulated visuals.

Overall, these were addressed well enough in rebuttal to justify acceptance, with the remaining caveat that conclusions should be stated with the right scope and limitations.

**Reviewer Concerns:**

See summary.

**Reviewer Scores:**

- KoNy: 8: positive, thinks concerns addressed, maintains rating.

- n8E9: 8: positive, strong on empirical value, asks for clarity + more analysis; authors respond with extra experiments.

- C7oU: 6: generally positive but wants clearer guidance + questions hyperparameter fairness; after rebuttal says “paper is in good shape” and keeps positive recommendation.

- ZXC9: 6: likes scale/clean comparison, but worries about simulation-only conclusions + novelty/positioning; does not visibly update score after rebuttal (thread says reviewer didn’t respond).

---

### Decision · Program_Chairs · 2026-01-26

Accept (Poster)